# Buddha image meditation is a potent predictor for mental health outcomes: A cross-sectional study among Thai high-school students

Justin DeMaranville[1], Tinakon Wongpakaran[1,2*], Nahathai Wongpakaran[1,2], Danny Wedding[1,3,4]

1 Mental Health Program, Multidisciplinary and Interdisciplinary School, Chiang Mai University, Chiang Mai, Thailand, 2 Department of Psychiatry, Faculty of Medicine, Chiang Mai University, Chiang Mai, Thailand, 3 Department of Clinical and Humanistic Psychology, Saybrook University, Pasadena, California, United States of America, 4 Department of Psychology, University of Missouri-Saint Louis, St. Louis, Missouri, United States of America

* tinakon.w@cmu.ac.th

## Abstract

### Purpose

Meditation has been demonstrated to benefit adolescent mental health. This research examined various meditation styles practiced in northern Thailand to determine which were associated with positive and negative mental health outcomes in adolescents.

### Population and methods

High school students who were 15–18 years old and who were enrolled in grades 10–12 in either secular or Buddhist Thai boarding schools were recruited following their school's willingness to participate.

They provided information about meditation styles and their practice frequency during the last month (i.e., breathing, kasina (color), Buddha image visualization, Manomay-iddhi, mindfulness, recollections, and vipassanā). The Rosenberg Self-Esteem Scale (RSES), Resilience Inventory (RI-9), Outcome Inventory-21 (OI-21), and Perceived Stress Scale (PSS-10) were completed. Multiple linear regression model analysis was used to identify the effects of meditation styles on mental health outcomes.

### Results

Among 443 participants, 390 were females (87.9%). The mean age was 16.35 ± 0.96 years. The three most common meditation styles practiced were breathing, Buddha image visualization, and mindfulness (46.5%, 26.2%, and 22.8%, respectively). Buddha image visualization was a significant predictor of RSES (B = 1.69, 95%CI = 0.77, 2.61), RI-9 (B = 2.95, 95%CI = 0.68, 2.95), OI-Anxiety (B = −2.38, 95%CI = −3.34, −1.41), OI-Depression (B = −1.94, 95%CI = −2.64, −1.24), and PSS-10 (B = −2.47,

**Data availability statement:** The authors confirm that all data underlying the findings are fully available without restriction. All relevant data are within the paper.

**Funding:** The authors received no specific funding for this work.

**Competing interests:** The authors have declared that no competing interests exist.

95%CI = −3.65, −1.28), whereas Manomayiddhi was a predictor of RI-9 (B = 2.47, 95%CI = 0.74, 2.47), OI-Anxiety (B = −2.32, 95%CI = −3.41, −1.23), OI-Depression (B = −1.53, 95%CI = −2.32, −0.74), and PSS-10 (B = −2.14, 95%CI = −3.46, −0.81). Breathing meditation predicted OI-Depression (B = −0.87, 95%CI = −1.45, −0.29). Daily meditation frequency was associated with the best mental health scores (p < 0.001).

## Conclusion

Buddha image visualization, Manomayiddhi, and breathing meditation were predictive of adolescents' mental health. A higher practice frequency is associated with positive mental health outcomes.

---

## Introduction

Research has documented that meditation is associated with and has positive effects on general health and mental health outcomes (e.g., stress and burnout) in both adolescents and adults [1,2] either before or during the COVID-19 pandemic [3,4]. However, research about meditation is often limited to well-known meditation styles, such as mindfulness, leaving uncertainty surrounding often practiced but less known styles. In addition to investigating the influence of various meditation styles on adolescent mental health, this study sought to understand how practice frequency influenced outcomes, with positive benefits reported amongst high-frequency meditators elsewhere [5]. Meditation is a widespread practice in Thailand that spans across all age groups. However, there is need for more comprehensive studies to accurately gauge the prevalence of meditation among high school students and the general population. It is a common ritual in schools to meditate and chant before class, and it's reasonable to expect that meditation practices are more prevalent and more structured in Buddhist schools compared to secular institutions. Approximately 17% of the general Thai population and as much as 58% of older residents actively participate in meditation, highlighting a significant number of practitioners with a variety of meditation styles [6,7].

Buddhist and secular meditation styles, such as mindfulness and loving-kindness, have shown positive outcomes for adolescent and adult self-esteem [8,9]. In addition to increased self-esteem, meditation has also been linked with increased resilience [10,11], an important quality for coping with stress, anxiety, and depression, which are commonly reported issues for adolescents during secondary education [12–14] and later adolescence [15]. Neurobiological research has found that meditation may influence mental health through changes in the brain's structure and in its functioning. Mindfulness meditation has association with increased activation in the prefrontal cortex and decreased activity in the amygdala [16,17]. These changes may support cognitive control and emotional reactivity, which may help with emotional regulation, resilience, and positive mental health outcomes during adolescence. Pre-pandemic research revealed parity in the mental health of students from various schools

(boarding and day schools); however, loneliness issues have been reported by boarding school students (15, 22). Studies conducted during the COVID-19 pandemic in Southeast Asian countries, including Thailand, have indicated that Thai adolescents reported significantly higher levels of stress, anxiety, and depression symptoms compared to their counterparts in other countries (17–20). Global estimates suggest that the prevalence of depression and anxiety doubled in response to the pandemic (21). School-based interventions to influence psychosocial outcomes have indicated moderate positive influences that can reduce the burden of school-related stressors [18]. Meditation practices and interventions, in and out of schools, have been shown to be effective treatments [19] for managing anxiety [20], depression [21,22], and stress [23]. Meditative interventions were also shown to be effective during the Covid-19 lockdown period [24]. There are many meditation styles practiced in Thailand and abroad, such as mindfulness and vipassanā. Buddha Image and Manomayiddhi are two styles not known by this study's authors to be practiced outside of Thailand nor previously researched. The remaining styles are common and discussed in Buddhist texts [25,26]. Some meditation styles are not difficult to learn or practice, whereas other techniques require guidance.

Breathing Meditation (Anapanasati) is a common meditative style in which a meditator directs his or her attention onto the sensation of the breadth at a chosen location on the body in order to focus attention. Research on breathing meditation has been demonstrated to reduce anxiety symptoms as opposed to a non-significant change in a control group [27]. Mindfulness Meditation is a well-known style internationally. The practice emphasizes mindful awareness of different experiences, such as of one's bodily sensations and feelings [28]. Mindfulness meditation courses and interventions have resulted in positive health results [29,30] demonstrated to be enduring [11]. The Ten Recollections (Anusti) refers to the contemplation of the following ten topics: the Buddha, dhamma (teachings), sangha (community of noble Buddhists), precepts (five moral virtues), generosity, the virtue of devas (angelic-like beings), mindfulness of breadth, body impurities, the virtue of nibbāna, and death [31]. The benefits of mindfulness of death have included reductions in death anxiety and increases in pro-social values [32]. Visualizing of Objects (Kasina) uses directed focus on an external and/or internal visualization of an image as a meditation object [25]. The ten types of kasina meditation are of the four primary elements (earth, water, fire, wind), the four colors (green, yellow/gold, red, white), bright light, and space [26]. Visual concentrative methods were found to have the strongest concentrative focus of focused attention types [33]. Buddha Image visualization focuses attention on an outline or detailed depiction of the Buddha. The reverence surrounding the Buddha as a historical and religious figure may help the meditator concentrate [34]. This meditation may include chanting the Buddha's name or visualizing the Buddha image in color, similar to kasina.

Vipassanā meditation is also referred to as insight meditation. Vipassanā (a Pali word meaning insight) can arise during any style of meditation. From Buddhism's perspective, vipassanā enables a person to overcome mental defilements. Vipassanā stems from observations of internal processes as they arise and disappear, seeing them as they are in terms of the three characteristics of suffering (impermanence, suffering, and non-self) and the four noble truths [35]. Manomayiddhi (mind-power meditation) combines breathing meditation, Buddha image visualization, recollections (i.e., of Buddha, dhamma, sangha, precepts, and death), and vipassanā meditation. A guide narrates (for novices) a sequence of steps, visualized as higher or lower realms or pure or impure symbols, which prompt meditators to heighten focus and discard mental impurities. Manomayiddhi is adapted from the training instructions in the Tipitaka (Buddhist cannons) [34,36,37]. The practitioner is encouraged to adhere to loving-kindness and five precepts (i.e., refraining from sexual misconduct, intoxicant use, killing, stealing, and lying).

This study seeks to determine the relationship between different meditation styles and mental health outcomes of adolescents in northern Thailand. Mindfulness meditation has received considerable research attention, with much less research investigating multiple meditation styles within the same population. Thai people practice a wide array of meditation types embedded in cultural contexts that influence their meditation practice. This study expands the conversation about meditation by accounting for mindfulness meditation as well as lesser-known styles such as Buddha Image and Manomayiddhi. Adolescent mental health in Thailand can then be assessed in consideration of the many meditation types

they may practice, as well as by accounting for their frequency of practice and sociodemographic influences. The authors investigated the seven aforementioned meditation styles to understand their associations with mental health. It was hypothesized that meditation would covary with self-esteem and resilience and that a negative association would exist between meditative styles and anxiety, depression, and perceived stress. However, no hypothesis was made regarding specific meditation styles and mental health outcomes. Practice frequency was hypothesized to influence mental health positively. We included age, sex, religion, secular or religious school, and socioeconomic status as control variables.

## Materials and methods

This research was an observation study. The study recruited 453 participants. The Inclusion criteria for the study were:

- Students who were 15 years old and older.

- Studying in grades 10–12.

- Studying in a Thai boarding school in Northern Thailand.

- All religious groups.

   The Exclusion criteria were students with special needs and students who were blind or deaf.

   The sample estimation was originally calculated for structural equation modeling (SEM) with a final sample of 453 participants. For this secondary data analysis, a sample size calculation for linear regression was determined by the medium effect size, significance level (alpha) of 0.05, power (beta) of 0.8. This yielded a number of at least 114 required for the analysis. The total sample of 453 was included. Boarding schools in northern Thailand were purposively selected to reflect similar socioeconomic status, the number of students, and the female-to-male ratio. Buddhist boarding schools were prioritized when contacting schools to ensure enough meditators participated. A research assistant contacted the schools by phone and email with information about the study. Two Buddhist schools in urban areas and three secular schools in urban and suburban areas from four provinces (Chiang Mai, Lamphun, Uthai Thani, Phitsanulok) participated. In the recruitment process, participating schools announced an information session that interested students could attend to learn about the research, with data collection occurring the following week. During this session, the informed consent was read and any questions were answered. Interested students were given approximately one week to speak with their families to ask for permission to participate. Since participation in this study was voluntary, not every student showed interest and chose to take part. The data collection period was between July and August 2021. From the two Buddhist schools, 179 and 57 students were recruited, and from the three secular schools, 145, 6, and 56 students were recruited. The questionnaires were administered both online and with paper depending upon the needs of the participating schools. The variation in students willing to participate was likely due to data collection occurring amidst a lockdown announcement, with many students opting to return home as opposed to sheltering at their boarding school. Written informed consent was obtained from participants and most of the participants' guardians. The ethics committee approved the schools to provide permission for students whose parents were unable to be reached. This study was approved by the Research Ethics Committee, Faculty of Medicine, Chiang Mai University, Thailand. Study (code 236/2021) and approved on 10 June 2021. A related study was published elsewhere [38].

### Instruments

Sociodemographic information about age, sex, religion, school type, and family monthly income was collected.

### Meditation evaluation questionnaire (MEQ)

The Meditation Evaluation Questionnaire was developed by Justin DeMaranville, Tinakon Wongpakaran, and Nahathai Wongpakaran to measure a variety of meditation styles and the frequency of a person's meditation practice. The

questions pertain to the meditator's practice over the last month. Questions include, 'How often do you meditate each week?' with responses ranging from (1) None, (2) 1–2 times, (3) 3–4 times, (4) 5–6 times, (5) every day. Different meditation styles are listed, and the participant is requested to indicate if he or she has used that style: vipassanā, Manomayiddhi, Buddha image visualization, color kasina, recollection (of death), anapanasati (breathing), or mindfulness.

### Rosenberg self-esteem scale (RSES)

The Rosenberg Self-Esteem Scale [39] is used internationally to measure self-esteem. The reliability and validity of the tool has had extensive testing in various languages and was found to be effective. A revised version of the tool was tested with Thai adolescents and found to be equally reliable but with better construct validity [40]. The RSES is brief and easily administered, utilizing a Likert scale with ten items answered on a four-point scale that ranges from strongly disagree (1) to strongly agree (4), with half of the items having positive wording and half negative wording. The total scores range from 10 to 40, with higher scores indicating higher self-esteem. The total scores for the Thai version range from 10 to 40, with higher scores indicating higher self-esteem. The scale was tested with this study's participants, resulting in a Cronbach's α of 0.80.

### Resilience inventory (RI-9)

This tool measures how well a person recovers after a setback or problem [41]. The tool has nine items that use a five-point Likert scale with total scores that range from 9 to 45. Higher scores indicate more resilience. The question responses range from "1—does not describe me at all" to "5—it describes me very well." The tool was tested with Thai university students with person reliability of 0.86 using Rasch analysis and a Cronbach's alpha value of 0.90. An additional use of the tool with medical students yielded an internal consistency of 0.86 [42]. The Cronbach's α for this study's population was 0.86.

### Outcome inventory (OI-21)

The Outcome Inventory is a 21-question self-rating tool to measure symptoms of anxiety, depression, somatization, and interpersonal problems. The questions use a Likert scale with values of (1) never, (2) rarely, (3) sometimes, (4) frequently, and (5) almost always. Subscale scores can be calculated as well as a total score that ranges from 0 to 84, with higher scores indicative of psychopathology. The Cronbach's alpha of a Thai clinical sample was found to be 0.92, with an anxiety subscale of 0.82 and a depression subscale of 0.87 [43]. The reliability for this study's anxiety subscale was α = 0.83 with a depression subscale α of 0.76.

### Perceived stress scale (PSS-10)

The Perceived Stress Scale has been used worldwide to assess a person's perception of stress during the last month [44]. The tool has ten items that use a five-point Likert scale with responses ranging from (0) never to (4) very often. Higher scores indicate that the participant has a greater perception of stress in his or her life. Six questions assess stress, while four are control questions. The Thai PSS-10 was found to have good internal validity (0.85) [45] and an acceptable Cronbach's α (0.77) in the current study.

### Statistical analysis

Overall, 3% of the data was found to be missing, and 344 out of 453 participants submitted surveys with missing data. Participant data were used if they completed 80% or more of the survey. Expectation Maximization (EM) imputation was used to fill in the randomly missing data. All analyses were performed using SPSS version 22. The descriptive data included frequencies, percentages, mean, and standard deviation (SD). The association between demographic data,

 

style of meditation, and mental health outcomes was assessed using analysis of variance (ANOVA) and Pearson's and biserial correlations. Multiple linear regression model analysis using the stepwise method for selective variables was used to determine the association along with other covariates including age, sex, and school type, as they were potential confounders. Stepwise regression is an iterative process of adding and removing variables based on their statistical significance and contribution to the model. The multiple linear regression equation for predicting mental health based on meditation types (ex. mindfulness, kasina, manomayiddhi) and covariates (age, sex, religion) can be represented as follows:

$$\text{Mental Health outcome} = \beta_0 + \beta_1 (\text{mindfulness}) + \beta_2 (\text{kasina}) + \beta_3 (\text{Manomayiddhi})$$
$$+ (\text{breathing}) + (\text{vipassana}) + (\text{Buddha image}) + (\text{recollection of death})$$
$$+ \beta_4 (\text{age}) + \beta_5 (\text{sex}) + \beta_6 (\text{religion}) + (\text{income}) + \varepsilon (\text{error})$$

Where: Mental health outcome is the dependent variable representing the individual's mental health status. Mindfulness, Kasina, and Manomayiddhi are the predictor variables representing different types of meditation. Age, sex, and religion are the covariates representing demographic and personal characteristics. $\beta_0$ is the intercept coefficient, while $\beta_1$, $\beta_2$, $\beta_3$, $\beta_4$, $\beta_5$, and $\beta_6$ are the regression coefficients for the respective predictors and covariates. The error term is represented as $\varepsilon$, capturing the variability in mental health that the predictors and covariates do not account for.

Unstandardized coefficients were used. Probability levels of $p < 0.05$ were considered statistically significant. The Bonferroni correction was used to accommodate seven independent variables (meditative styles) [.05/7]; therefore, adjusted p-values less than 0.007 were considered statistically significant.

## Results

In total, 443 responses were included in the analysis out of 453. Ten outliers were identified in the regression analysis and excluded from the final data. Table 1 lists the participants' sociodemographic characteristics.

For clinical data, the participants' meditative styles and weekly meditation practice frequency distributions are represented in pie charts. Figure 1 highlights the variety of meditation styles practiced by this Thai teenage boarding school student population. It was found that 132 participants practiced more than one meditation style last month.

Fig 2 highlights the weekly meditation practice of the participants during the last month.

**Table 1. Sociodemographic Characteristics of the Participants.**

| Variables (n = 443) | N or mean ± SD | % |
|---|---|---|
| Age | 16.35 ± 0.96 | – |
| Sex, female | 390 | 88 |
| Sex, male | 53 | 12 |
| School type | | |
| Buddhist | 236 | 53.4 |
| Secular | 207 | 46.6 |
| Religion | | |
| Buddhism | 396 | 89.4 |
| Non-Buddhist | 47 | 10.6 |
| Income (month) | | |
| Family income—less than USD 295* | 237 | 53.9 |
| Family income—USD 296 and higher | 206 | 46.1 |

* 1 USD = 32 THB (exchange rate at the time of the study), SD = standard deviation

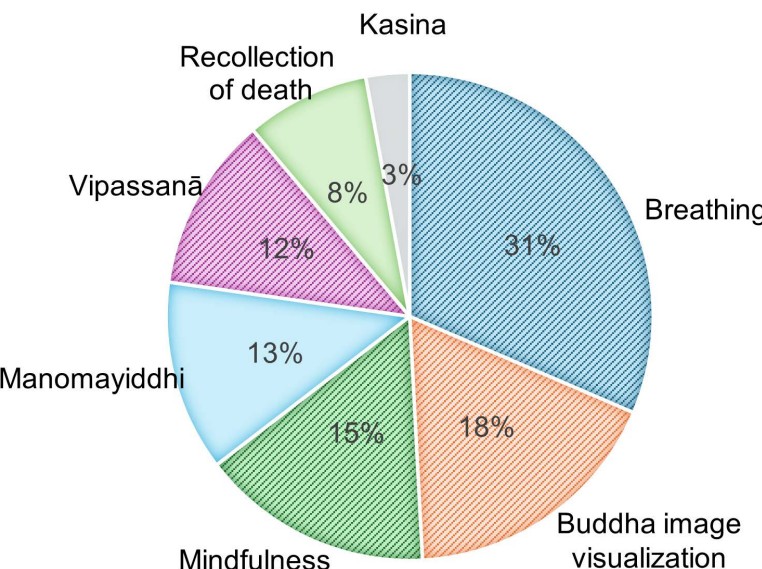

**Fig 1. Meditation Styles of Thai Teenagers.** The most common meditative styles reported as having been practiced in the last month by this population were breathing meditation, Buddha image visualization, and mindfulness meditation.

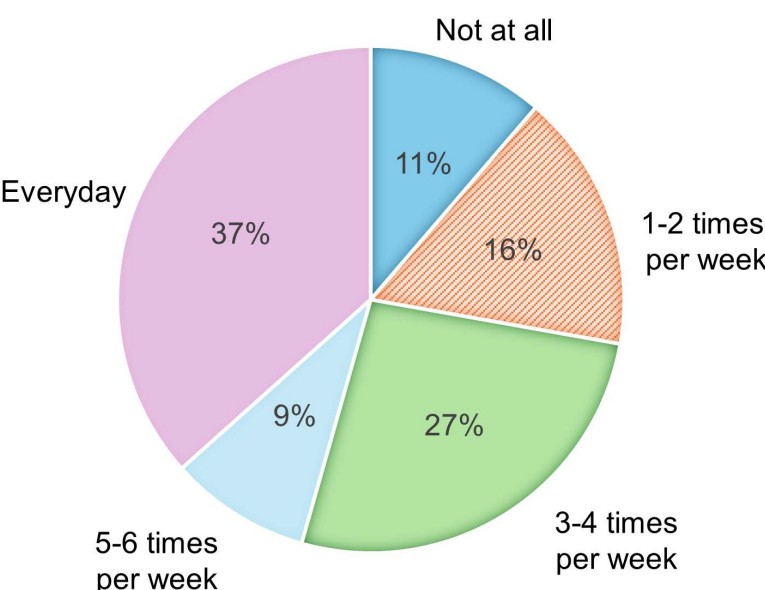

**Fig 2. Meditation Frequency of Thai Teenagers.** On average, the participants meditated approximately 3-4 times per week (2.43 ± 1.41).

Table 2 is a zero-order correlation table with demographic data, mental health outcomes, and meditation practices. The meditative styles were positively related to self-esteem and resilience, and negative relationships were found with anxiety, depression, and stress. Vipassanā, mindfulness, kasina (color), and breathing meditations produced small effect sizes and mostly non-significant associations. Buddha image, manomayiddhi, and recollection of death were mostly significant and had larger effect sizes. Moderate effect sizes between school type (Buddhist) and manomayiddhi (.44) and Buddha image

**Table 2. Zero-Order Correlation Table of Demographic Information and Mental Health.**

| Variable | N (%) or Mean±SD | 1 | 2 | 3 | 4 | 5 | 6 | 7 | 8 | 9 | 10 | 11 | 12 | 13 | 14 | 15 | 16 |
|---|---|---|---|---|---|---|---|---|---|---|---|---|---|---|---|---|---|
| 1.Age | 16.35±0.96 | − | .039 | −.123** | −.110* | −.158** | .072 | .083 | −.099* | −.047 | −.027 | −.094* | −.086 | −.133** | −.055 | .045 | −.015 |
| 2.Sex (Female) | 390 (87.9) | | − | .192** | .059 | .214** | .050 | .086 | −.069 | −.122* | −.064 | .164** | .304** | .176** | .059 | .005 | .139** |
| 3.School Type (Buddhist) | 236 (53.4%) | | | − | .045 | .247** | .150** | .283** | −.205** | −.248** | −.212** | .390** | .438** | .506** | .087 | −.125** | .142** |
| 4. Religion (Buddhist) | 396 (89.4%) | | | | − | .189** | .128** | .126** | −.024 | −.187** | −.119* | .262** | .165** | .155** | .073 | .130** | −.051 |
| 5.Monthly Income (≤5000 Baht) | 237 (53.9%) | | | | | − | .182** | .191** | −.132** | −.199** | −.202** | .261** | .295** | .299** | .160** | .056 | .151** |
| 6.Self-esteem (RSES) | 30.57 (4.14) | | | | | | − | .587** | −.504** | −.557** | −.612** | .370** | .206** | .239** | .014 | .061 | .104* |
| 7.Resilience (RI) | 33.82 (5.84) | | | | | | | − | −.408** | −.471** | −.484** | .408** | .280** | .319** | .092 | .050 | .164** |
| 8. Perceived Stress (PSS) | 23.94 (5.43) | | | | | | | | − | .642** | .556** | −.265** | −.211** | −.235** | −.092 | −.074 | −.111* |
| 9. Anxiety (OI-Anxiety) | 7.67 (4.59) | | | | | | | | | − | .776** | −.448** | −.288** | −.302** | −.046 | −.062 | −.073 |
| 10. Depression (OI-Depression) | 3.75 (3.34) | | | | | | | | | | − | −.437** | −.267** | −.317** | −.034 | −.130** | −.045 |
| 11. Meditation Frequency | 2.43 (1.41) | | | | | | | | | | | − | .420** | .485** | .141** | .126** | .170** |
| 12. Manomayiddhi | 83 (18.7%) | | | | | | | | | | | | − | .346** | .241** | .005 | .263** |
| 13. Buddha Image | 116 (26.2%) | | | | | | | | | | | | | − | .178** | .001 | .249** |
| 14. Kasina (color) | 4.3 (19%) | | | | | | | | | | | | | | − | .138** | .330** |
| 15. Breathing | 206 (46.5%) | | | | | | | | | | | | | | | − | .081 |
| 16. Recollection of death | 54 (12.2%) | | | | | | | | | | | | | | | | − |

SD = standard deviation.

**Bolded** results indicate significance at *p<0.05, **p<0.01.

(.51) meditations were found, indicating these styles were mostly practiced in Buddhist schools. Participants meditated approximately 3–4 times per week (2.43±1.41). Strong effect sizes were found between meditation frequency with Buddha Image (.485, p<0.01) and Manomayiddhi (.42, p<0.01).

Table 3 is an ANOVA comparison of the weekly meditation frequency and self-esteem, resilience, perceived stress, anxiety, and depression scores of this population. The p-values for all comparisons document that significant differences exist between meditation frequencies and self-esteem ($F=17.86$, $p<0.001$), resilience ($F=22.74$, $p<0.001$), perceived stress ($F=10.55$, $p<0.001$), anxiety ($F=28.08$, $p<0.001$), and depression ($F=27.21$, $p<0.001$). A higher frequency of meditation practice predicted higher self-esteem and resilience scores. Lower frequency of meditation practice predicted symptoms of anxiety, depression, and stress.

Table 4 is a multiple linear regression analysis of positive mental health outcomes, having used the stepwise method for selecting variables. Both the self-esteem and resilience models used four iterations to arrive at the final model. These data revealed that Buddha image and Manomayiddhi significantly predicted self-esteem ($\beta=.180$, $p<0.001$, $\beta=.121$, $p=.015$) and resilience ($\beta=.223$, $p<0.001$, $\beta=.165$, $p=0.001$).

Table 5 accounts for the influence of meditative styles on negative mental health outcomes. Regarding the stepwise method used, four iterations were used in the anxiety model, with three iterations used in the depression and perceived stress models. The results revealed that Buddha image and Manomayiddhi were significant predictors for anxiety symptoms ($\beta=−.288$, $p<0.001$, $\beta=−.197$, $p=0.001$, respectively) and perceived stress ($\beta=−.200$, $p<0.001$, $\beta=−.154$, $p=0.002$, respectively). For depression, visualization of the Buddha image, Manomayiddhi, and breathing meditation were significant predictors ($\beta=−.255$, $p<0.001$, $\beta=−.179$, $p<0.001$, $\beta=−.129$, $p=0.004$, respectively).

**Table 3. Mental Health Scores According to Weekly Meditation Frequency During the Last Month.**

| Variables | Weekly Meditation Frequency | N | Mean Scores | SD | 95% C.I. | F (4,438) | p-value |
|---|---|---|---|---|---|---|---|
| Self-esteem | None | 50 | 28.43 | 4.61 | (27.12, 29.74) | 17.86 | <0.001 |
| | 1-2 times | 73 | 28.78 | 3.99 | (27.85, 29.71) | | |
| | 3-4 times | 118 | 29.85 | 3.86 | (29.15, 30.55) | | |
| | 5-6 times | 40 | 31.47 | 4.11 | (30.15, 32.78) | | |
| | Everyday | 162 | 32.34 | 3.46 | (31.8, 32.88) | | |
| Resilience | None | 50 | 30.28 | 5.34 | (28.76, 31.8) | 22.74 | <0.001 |
| | 1-2 times | 73 | 31.63 | 5.32 | (30.39, 32.87) | | |
| | 3-4 times | 118 | 32.42 | 5.06 | (31.5, 33.34) | | |
| | 5-6 times | 40 | 34.60 | 5.74 | (32.77, 36.44) | | |
| | Everyday | 162 | 36.72 | 5.44 | (35.87, 37.56) | | |
| Perceived Stress | None | 50 | 25.26 | 4.37 | (24.02, 26.5) | 10.55 | <0.001 |
| | 1-2 times | 73 | 25.92 | 4.12 | (24.96, 26.88) | | |
| | 3-4 times | 118 | 25.11 | 5.53 | (24.1, 26.12) | | |
| | 5-6 times | 40 | 21.93 | 6.08 | (19.98, 23.87) | | |
| | Everyday | 162 | 22.28 | 5.40 | (21.44, 23.12) | | |
| Anxiety | None | 50 | 10.68 | 3.90 | (9.57, 11.79) | 28.09 | <0.001 |
| | 1-2 times | 73 | 10.18 | 4.10 | (9.22, 11.14) | | |
| | 3-4 times | 118 | 8.41 | 4.02 | (7.68, 9.14) | | |
| | 5-6 times | 40 | 6.52 | 4.75 | (5, 8.04) | | |
| | Everyday | 162 | 5.35 | 4.09 | (4.72, 5.98) | | |
| Depression | None | 50 | 5.81 | 3.47 | (4.82, 6.79) | 27.21 | <0.001 |
| | 1-2 times | 73 | 5.81 | 3.23 | (5.05, 6.56) | | |
| | 3-4 times | 118 | 4.03 | 3.10 | (3.46, 4.59) | | |
| | 5-6 times | 40 | 3.38 | 3.47 | (2.27, 4.49) | | |
| | Every day | 162 | 2.09 | 2.53 | (1.69, 2.48) | | |

SD = standard deviation, CI = confidence interval

**Table 4. Regression Analysis Results for Self-Esteem and Resilience.**

| Model | B | SE | β |
|---|---|---|---|
| Self-Esteem | | | |
| (Constant) | 20.721 (14.23, 27.22) *** | 3.30 | |
| Buddha Image | 1.693 (0.77, 2.61) *** | 0.47 | 0.180 |
| Manomayiddhi | 1.286 (0.25, 2.32) * | 0.53 | 0.121 |
| Age | 0.534 (0.14, 0.93) ** | 0.20 | 0.124 |
| Income | 0.927 (0.13, 1.73) * | 0.41 | 0.112 |
| Resilience | | | |
| (Constant) | 17.84 (9.10,26.60) *** | 4.46 | |
| Buddha image | 2.953 (1.61, 4.30) *** | 0.68 | 0.223 |
| Manomayiddhi | 2.468 (1.02, 3.92) ** | 0.74 | 0.165 |
| Age | 0.858 (0.32, 1.38) ** | 0.27 | 0.141 |
| School Type | 1.345 (0.11, 2.57) * | 0.63 | 0.115 |

B = Unstandardized Coefficients, SE = standard error, β = Standardized Coefficients.

*p < 0.05, **p < 0.01, ***p < 0.001.

**Table 5. Regression Analysis for Anxiety, Depression, and Perceived Stress.**

| Model | B | SE | β |
|---|---|---|---|
| Anxiety | | | |
| (Constant) | 18.969 (11.88, 26.06) *** | 3.60 | |
| Buddha Image | −2.375 (−3.34, −1.41) *** | 0.49 | −0.228 |
| Manomayiddhi | −2.321 (−3.41, −1.23) *** | 0.55 | −0.197 |
| Religion | −1.954 (−3.27, −0.64) ** | 0.67 | −0.131 |
| Age | −0.520 (−0.94, −0.10) * | 0.21 | −0.109 |
| Depression | | | |
| (Constant) | 4.950 (4.51, 5.39) *** | 0.22 | |
| Buddha Image | −1.939 (−2.64, −1.24) *** | 0.36 | −0.255 |
| Manomayiddhi | −1.529 (−2.32, −0.74) *** | 0.40 | −0.179 |
| Breathing | −0.866 (−1.45, −0.29) ** | 0.30 | −0.129 |
| Perceived Stress | | | |
| (Constant) | 37.829 (29.42, 46.24) *** | 4.28 | |
| Buddha image | −2.465 (−3.65, −1.28) *** | 0.60 | −0.200 |
| Manomayiddhi | −2.138 (−3.46, −0.81) ** | 0.67 | −0.154 |
| Age | −0.786 (−1.30, −0.28) ** | 0.26 | −0.139 |

B = Unstandardized Coefficients, β = Standardized Coefficients, SE = standard error.

*p < 0.05, **p < 0.01, ***p < 0.001.

## Discussion

The purpose of the current study was to contribute to the discussion surrounding meditation styles and mental health outcomes. Consistent with other research, meditation in this study was positively associated with self-esteem [46] and resilience [11], whereas it correlated negatively with anxiety, depression, and stress [22,23,47–49].

Color kasina, vipassanā, and mindfulness meditations produced small effect sizes and had mostly non-significant associations; however, color kasina meditation neared significance with self-esteem and resilience. The small, significant correlation between kasina and practice frequency may explain the trend toward significance as regular meditation practice positively influenced mental health in other research [5,50]. The small number of kasina practitioners highlights that it is not a common practice, and this may have influenced the quality of the meditation and instruction, contributing to its non-significance. Unexpectedly, mindfulness was non-significantly associated with mental health in contrast to the often-reported findings about mindfulness meditation effectiveness [9,51,52].

Among the seven types of meditation analyzed, only Buddha image visualization, Manomayiddhi, and breathing meditation proved to be predictors of mental health. For the Buddha image, the researchers hypothesize that visualized features of the Buddha, such as facial expression or cloth details, reflect the meditator's state of mind. Visualizing the Buddha's features as clear is related to the meditator's focus. The reverence surrounding the Buddha as a religious figure may help with generating concentration on the image, particularly among Buddhists. The mechanism behind Buddha image visualization effectiveness has not yet been fully understood. It may be that the visualization of the Buddha affects the mental well-being of Thai people positively, with previous research about mental imagery and guided imagery indicating reductions in depression and suicidality [53,54].

Manomayiddhi's efficacy may be due to the variety of meditation styles practiced using this approach. Initially, either Buddha Image or breathing meditation is used to create calm and focus. Afterward, a brief chanting is used to formally begin Manomayiddhi. The meditator begins by visualizing the Buddha image until the meditator's level of concentration is sufficient for practicing recollections, such as death meditation. Recollections of death have been found to reduce

death anxiety and increase pro-social values [32]. The process through these meditative styles generates vipassanā (insight) into Buddhist teachings [34]. As these styles are transitioned through during Manomayiddhi, a meditator's focus changes in relation to the body. Breathing meditation has been documented to have higher body orientation and activation, whereas chanting and contemplative meditative styles are less focused on the body [33]. This process may affect the meditator with the soothing benefits of breadth meditation or Buddha image visualization and with the insight gained from contemplation of the recollections. However, the integration of the aforementioned types of meditation into Manomayiddhi raises questions about why recollections, vipassanā, mindfulness, and kasina were not significant predictors in the regression models. Some explanation for these styles' non-significant results may be explained by consideration of the quality of instruction and frequency of practice.

In this sample, correlation results highlight that Buddha image and Manomayiddhi are practiced predominately at Buddhist boarding schools as opposed to secular schools, highlighting how school types differ in relation to meditation practice. Buddhist boarding schools are often connected to a Buddhist temple that designates monks to instruct the students, a factor which can influence the quality of meditation instruction. Secular schools lack temple affiliation and may not emphasize meditation as part of the curriculum. Additionally, regional differences in training styles can elevate the practice of certain meditation types over others. In the case of Manomayiddhi, meditators will generally use Buddha image visualization; however not all Buddha image meditators practice Manomayiddhi. Buddha image visualization can be a standalone technique, but it is often practiced for focus prior to Manomayiddhi. This is similar to meditators who begin with breathing meditation before they practice vipassana. This interaction of meditation types suggests one type may influence another, and this is important to account for in controlled studies. It can be the case within religious schools in Thailand that Manomayiddhi is taught with a guide for novice meditators, which can elevate the quality of the practice beyond unguided meditation. It also can mean that Buddha image meditation may be guided if used prior to guided Manomayiddhi. It may be that the instruction benefits Buddha image practice, and differences in outcome might be found between guided and non-guided Buddha image and Manomayiddhi meditators.

Beyond school type and instruction quality, another aspect of Manomayiddhi practice is that training in Manomayiddhi usually includes encouragement to follow the first five (or eight) Buddhist precepts. Observing the five precepts is to accept and adopt the belief that killing, stealing, sexual misconduct, lying, and substance use are impediments to concentration and self-regulation. Precept observances are impactful regulatory behaviors for adolescents, previously demonstrated to buffer the effect of perceived stress on depression [55] and to have a mediating effect between parental attachment and resilience in adolescents [38]. Observing those five behaviors in tandem with Manomayiddhi practice also means that frequent meditators of Buddha image, who also practice Manomayiddhi, benefit from these five Buddhist precepts. The precepts are an additional factor to include in explaining how Buddha image visualization and Manomayiddhi influence this population's positive and negative mental health outcomes as opposed to the other meditative styles found to be non-significant.

The third predictor of mental health in the regression models was breathing meditation, which was significant only in the depression model. This particular meditation was found to be practiced by nearly half of this study's participants, which was the largest number of practitioners for any single meditation style. Related studies found breathing meditation reduced anxiety and stress [27,54]. Breathing meditation includes voluntary regulation of the breadth, a process that influences psychophysiological changes in parasympathetic activity, cardiac function, and increased electroencephalography alpha power and decreased theta power, which were found to reduce depression [56]. The positive association found between breathing meditation and practice frequency highlights that common practice is a constant influence amongst significant meditation styles in this study.

Buddha image and Manomayiddhi, the two most potent predictors of mental health outcomes, correlated most highly with practice frequency. These data also indicate daily or near daily practice positively influenced all mental health outcomes the most. In other research, a daily meditation practice reduced meditator stress and anxiety symptoms [57].

Another study showed that six days-a-week at-home mindfulness interventions were shown to reduce anxiety, depression, and stress effectively [58]. Three or more days a week of meditation among people with major depression indicated a reduced risk of relapse into major depression by almost half when compared with less than three days per week of meditation at a 12-month follow-up [50]. These findings highlight that the assessment of meditation frequency is an important parameter when assessing the impact of meditation.

To the best of our knowledge, this study is the first to assess meditation styles in Thailand and their association with mental health outcomes. Nevertheless, this study has limitations. Schools and students were recruited for the study without the use of probability sampling techniques, such as with the purposive selection of two Buddhist boarding schools. Participants also joined following announcements at participating schools, rather than by randomly selecting participants from those schools. This limits how these results can be generalized. Regarding meditation practice, the participants were able to select more than one meditation practice without indicating why a given meditation was preferred over others, how frequently each meditation was practiced, and if meditation types were combined. For example, breathing meditation may be practiced for five minutes before vipassana meditation begins. That information, along with other qualitative information, would help in evaluating the accessibility, difficulty, and benefits of these meditation types. Regarding kasina meditation, the questionnaire listed color kasina, as this is more common than kasina with elements, yet specifying color may result in participants not selecting this type. Data were not collected on previous meditation experiences, leaving uncertainty regarding the long-term influence of meditation in this population. No objective measure for validation of each method can be confirmed. Likewise, we do not have hard evidence to prove our hypothesis, (e.g., electroencephalography, brainwave analysis). The generalizability of our results outside of boarding school students is limited. Future research should include longitudinal data to confirm the benefits of these meditative styles on mental health outcomes. A randomized controlled study of these particular styles of meditation is warranted.

## Conclusions

This research demonstrates that meditation type yields varied mental health results among Thai boarding school adolescents. While mindfulness, kasina, vipassanā, and recollections meditations were not significant, Buddha image visualization, Manomayiddhi, and breathing meditations were found to be effective and significant predictors using both positive and negative indices of mental health. As anticipated, frequent meditation practice predicted better mental health outcomes. This research may be the first to have assessed how Buddha image and Manomayiddhi influence mental health, demonstrating gaps in meditation techniques that warrant exploration in future studies. This research also highlights the complexity of observational studies about meditation, particularly the use of mixing meditation techniques. Future research on meditation types should include practice frequency, detailed documentation of meditator experiences, as well as the factors associated with meditation's effects on mental health issues.

## Acknowledgments

The authors thank the school officials who accommodated our research team. We also thank our diligent research assistant.

## Author contributions

**Conceptualization:** Justin DeMaranville, Tinakon Wongpakaran, Nahathai Wongpakaran, Danny Wedding.

**Data curation:** Justin DeMaranville, Tinakon Wongpakaran.

**Formal analysis:** Justin DeMaranville, Tinakon Wongpakaran, Nahathai Wongpakaran.

**Investigation:** Justin DeMaranville, Tinakon Wongpakaran, Nahathai Wongpakaran.

**Methodology:** Justin DeMaranville, Tinakon Wongpakaran, Nahathai Wongpakaran, Danny Wedding.

**Project administration:** Tinakon Wongpakaran, Nahathai Wongpakaran.

**Resources:** Tinakon Wongpakaran, Nahathai Wongpakaran, Danny Wedding.

**Software:** Tinakon Wongpakaran.

**Supervision:** Tinakon Wongpakaran, Nahathai Wongpakaran, Danny Wedding.

**Validation:** Tinakon Wongpakaran, Nahathai Wongpakaran.

**Writing – original draft:** Justin DeMaranville.

**Writing – review & editing:** Tinakon Wongpakaran, Nahathai Wongpakaran, Danny Wedding.

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
