## [Decision Letter · Decision Letter 0]

PONE-D-23-03807Buddha image meditation strongly predicts mental health outcomes: a cross-sectional study among high-school students.PLOS ONE

Dear Dr. Wongpakaran,

Thank you for submitting your manuscript to PLOS ONE. After careful consideration, we feel that it has merit but does not fully meet PLOS ONE’s publication criteria as it currently stands. Therefore, we invite you to submit a revised version of the manuscript that addresses the points raised during the review process.

We look forward to receiving your revised manuscript.

Kind regards,

Sally Mohammed Farghaly

Academic Editor

PLOS ONE

Journal Requirements:

a) Did participants provide their written or verbal informed consent to participate in this study?

Reviewers' comments:

Reviewer's Responses to Questions

**Comments to the Author**

1. Is the manuscript technically sound, and do the data support the conclusions?

Reviewer #1: Partly

Reviewer #2: No

2. Has the statistical analysis been performed appropriately and rigorously? 

Reviewer #1: No

Reviewer #2: Yes

3. Have the authors made all data underlying the findings in their manuscript fully available?

Reviewer #1: Yes

Reviewer #2: No

4. Is the manuscript presented in an intelligible fashion and written in standard English?

Reviewer #1: Yes

Reviewer #2: No

5. Review Comments to the Author

Reviewer #1: 1. Experiments, statistics, and other analyses are performed to a high technical standard and are described in sufficient detail.

a. More details on recruitment and data collection are needed (e.g., when was data collected, paper or electronic surveys, completed in school or at home, etc.).

b. Under Materials and Methods (p. 5) – how was quality of meditation practice measured using the MEQ? Only questions regarding meditation styles and frequency are described.

c. Under Materials and Methods (p. 5-6) – for consistency, please include Cronbach’s alphas for all instruments used (it was reported for RI-9 and PSS-10 but not the other measures).

d. Please provide more information on how imputation was performed, i.e., what data were used to impute the missing values.

e. There is no mention of ANOVA in the Statistical Analysis section (p. 7), but results from an ANOVA are later described under Results and Discussion (p. 10). Please include a description of the ANOVA performed in the Statistical Analysis section.

f. Under Results and Discussion (p. 7) it says “Ten outliers were excluded from the final data” – what criteria were used to determine outliers? This should be noted in the Statistical Analysis section.

g. Why were regressions not estimated for the other meditation types, i.e., those not included in the paper? Why were some demographic variables not included in the regressions?

2. Conclusions are presented in an appropriate fashion and are supported by the data.

a. “A small correlation between kasina and practice frequency may explain the trend” (p. 12) – please explain this further. Has this been demonstrated in past research?

b. “Mindfulness meditation practice is common in Thailand, but it is not often practiced by adolescents in northern Thailand” (p. 13) – please explain this further. Has this been demonstrated in past research?

c. “Among all styles of meditation, only Buddha image visualization, Manomayiddhi, and breathing meditation have proved to be predictors of mental health” (p. 13) – in the current sample yes, but the way it is worded makes it sound like you are referring to this generally, which is not accurate.

d. Second paragraph on p. 13 – it would make more sense to discuss the results for each style in the order they are listed in that first sentence (“Among all styles of meditation, only Buddha image visualization, Manomayiddhi, and breathing meditation have proved to be predictors of mental health”). Consider reorganizing this section so that Buddha visualization is discussed first, then Manomayiddhi, and then breathing meditation.

e. Last paragraph on p. 14 – please elaborate on why Buddha image visualization and Manomayiddhi are mostly practiced at Buddhist boarding schools. You make this statement, but the following sentences don’t necessarily connect back to this point.

f. “Data regarding negative experiences during meditation were not collected, a relevant consideration as perceived stress scores indicated infrequent or everyday meditation may be a stressor” (p. 16) – please elaborate on this. Has this been seen in previous research? Further discussion on these findings is warranted.

g. In the Conclusion section (p. 16), please provide additional context for these results (e.g., “This research demonstrates that many types of meditation yield varied results among adolescents in Thailand”).

3. The article is presented in an intelligible fashion and is written in standard English.

a. In general, the writing can be difficult to follow and switches between past and present tense.

4. The research meets all applicable standards for the ethics of experimentation and research integrity.

a. Yes, the authors indicate the study received approval from an ethics committee and informed consent was obtained from participants.

5. The article adheres to appropriate reporting guidelines and community standards for data availability.

a. Under Data Availability, the authors state that some restrictions will apply but then below state that the data underlying the findings are fully available without restriction. Please clarify.

Other notes:

6. Introduction

a. I think it would be beneficial to elaborate on mental health issues among adolescents in Asia during/after COVID-19 lockdowns, as well as any past research on meditation-based interventions in schools in Thailand.

b. Has past research examined the mental health benefits of Kasina, Buddha image visualization, and Manomayiddhi? If so, please include that in the introduction.

7. Overall

a. There seems to be a large overlap in the content of these different meditation styles, e.g. Buddha image is part of Manomayiddhi, so these results do not help us understand what components of Manomayiddhi may be helpful.

b. There is no discussion of why some styles might not be associated with better mental health.

Reviewer #2: Thank you for giving me the opportunity to read and comment on this paper. I find the topic very interesting. I thank the authors for their efforts at these times. Having said that, I suggested some comments which should be addressed.

1. The abstract section needs more changes, as the main results did not well articulate. Pay attention to reporting the main results of the study, as the results in the abstract do not convey the main findings of your study. The background paragraph in "Abstract" must be deleted and changed. The background section should be the shortest part of the abstract and should very briefly outline: What is already known about the subject, is related to the paper in question. What is not known about the subject and hence what the study intended to examine??

2. 3. The introduction (background) here is not adequate. The background of your study did not provide context to the information discussed throughout the research paper. The background should include both important and relevant studies on meditation and it's associated with general health and mental health outcomes. This is particularly important to support your paper.

3. Please report the full questionnaire used in this study.

4. What is the non-response rate of the study? Moving forward, all reviewers will likely ask for a response rate. The study should state 1) clearly how participant recruitment was performed - e.g. did study personnel reach out to participants, or some other mechanism?, e.g. were email lists used? 2) how many participants received the survey? 3) of those who received the survey, how many participants responded- completely (plus how do you define complete) or partially? 4) for your analyses, did you include complete or partial responses?

5. Please add about how you try to maintain the correctness and accuracy of the data during data collection.

6. Please add how sample size was calculated.

7. Please report results and discussion in separate sections. The discussion was not adequate and need to mention related studies to provide readers with appropriate interpretations. Pay attention to mentioning more related studies with respect to your study. The discussion section is not backed up by a sufficient literature review. Recent references can be added to strengthen the argument of the impact of image meditation on mental health outcomes.

6. PLOS authors have the option to publish the peer review history of their article (what does this mean? ). If published, this will include your full peer review and any attached files.

**Do you want your identity to be public for this peer review?** For information about this choice, including consent withdrawal, please see our Privacy Policy .

Reviewer #1: No

Reviewer #2: No

---

## [Author Response · Author response to Decision Letter 1]

8 May 2023

Dear editor and reviewers,

We appreciate your valuable comments and suggestions. Please see below our point-by-point responses.

Response. We have checked our manuscript accordingly.

a) Did participants provide their written or verbal informed consent to participate in this study?

Response: Participants and guardians provided written consent. This was amended in the Materials and methods section and reads, “Written informed consent was obtained from participants and most of the participant’s guardians. The school administrators provided written consent for participants whose parents were unable to be reached.”

Response. We have attached the data file to this submission.

Response: the title in the manuscript has been updated in line with the online submission form. “Buddha image meditation is a potent predictor for mental health outcomes: a cross-sectional study among high-school students.”

Response: the full name of the ethics committee and that the participants provided written consent was included in the Materials and methods section and reads, “Written informed consent was obtained from participants and most of the participant’s guardians… This study was approved by the Research Ethics Committee, Faculty of Medicine, Chiang Mai University, Thailand. The study code is 236/2021 and the date of approval was 10 June 2021.”

Reviewers' comments:

Reviewer's Responses to Questions

Comments to the Author

Reviewer #1: 1. Experiments, statistics, and other analyses are performed to a high technical standard and are described in sufficient detail.

a. More details on recruitment and data collection are needed (e.g., when was data collected, paper or electronic surveys, completed in school or at home, etc.).

Response: Data collection is specified in Materials and methods section - between July and Aug 2021. Information was provided about survey type and location of completion in Methods and materials section, “Data collection was conducted with both electronic and paper surveys when the students were at their schools.”

b. Under Materials and Methods (p. 5) – how was quality of meditation practice measured using the MEQ? Only questions regarding meditation styles and frequency are described.

Response: The remark about quality was deleted.

c. Under Materials and Methods (p. 5-6) – for consistency, please include Cronbach’s alphas for all instruments used (it was reported for RI-9 and PSS-10 but not the other measures).

Response: Cronbach’s alphas were included in Instruments section for RSES α=0.80, OI-anxiety subscale α=0.83, and OI-depression subscale α=0.76.

d. Please provide more information on how imputation was performed, i.e., what data were used to impute the missing values.

Response: Expectation Maximization (EM) imputation was used to fill in the randomly missing data.

e. There is no mention of ANOVA in the Statistical Analysis section (p. 7), but results from an ANOVA are later described under Results and Discussion (p. 10). Please include a description of the ANOVA performed in the Statistical Analysis section.

Response: the section Statistical Analysis has been edited to include mention of ANOVA. “The association between demographic data, style of meditation, and mental health outcomes was assessed using analysis of variance (ANOVA) and Pearson’s and biserial correlations.”

f. Under Results and Discussion (p. 7) it says “Ten outliers were excluded from the final data” – what criteria were used to determine outliers? This should be noted in the Statistical Analysis section.

Response: Ten outliers were identified in the regression analysis; the residual and distance values were calculated. The cases suggesting outliers were from the final data.

g. Why were regressions not estimated for the other meditation types, i.e., those not included in the paper? Why were some demographic variables not included in the regressions?

Response: The other meditation styles were included in the stepwise regression, yet they were found to be non-significant at every step in the process.

2. Conclusions are presented in an appropriate fashion and are supported by the data.

a. “A small correlation between kasina and practice frequency may explain the trend” (p. 12) – please explain this further. Has this been demonstrated in past research?

Response: This remark was expanded and a citation provided that reads, “A small, significant correlation between kasina and practice frequency (Table 2) may explain the trend toward significance as regular meditation practice was found to positively influence mental health in other research {Basso, 2019, Brief`, Daily Meditation Enhances Attention`, Memory`, Mood`, and Emotional Regulation in Non-Experienced Meditators}.”

b. “Mindfulness meditation practice is common in Thailand, but it is not often practiced by adolescents in northern Thailand” (p. 13) – please explain this further. Has this been demonstrated in past research?

Response: This sentence was deleted.

c. “Among all styles of meditation, only Buddha image visualization, Manomayiddhi, and breathing meditation have proved to be predictors of mental health” (p. 13) – in the current sample yes, but the way it is worded makes it sound like you are referring to this generally, which is not accurate.

Response: This sentence was edited to be more clear, “Among seven types of meditation analyzed for, only Buddha image visualization, Manomayiddhi, and breathing meditation proved to be predictors of mental health.”

d. Second paragraph on p. 13 – it would make more sense to discuss the results for each style in the order they are listed in that first sentence (“Among all styles of meditation, only Buddha image visualization, Manomayiddhi, and breathing meditation have proved to be predictors of mental health”). Consider reorganizing this section so that Buddha visualization is discussed first, then Manomayiddhi, and then breathing meditation.

Response: this recommendation is appreciated, and it has been implemented.

e. Last paragraph on p. 14 – please elaborate on why Buddha image visualization and Manomayiddhi are mostly practiced at Buddhist boarding schools. You make this statement, but the following sentences don’t necessarily connect back to this point.

Response: Indeed, more information was provided regarding this important point, “In this sample, correlation results highlight that Buddha image and manomayiddhi are practiced more frequently at Buddhist boarding schools and not at secular schools, highlighting how school types differ in relation to meditation practice. School type was also found to be a significant influence in the resilience regression model. Buddhist boarding schools are often connected to a Buddhist temple that designates monks to instruct the students, a factor which can influence the quality of meditation instruction. Secular schools can lack temple affiliation and may not emphasize meditation as part of the curriculum. Additionally, regional differences in training styles can elevate the practice of certain meditation types over others.”

f. “Data regarding negative experiences during meditation were not collected, a relevant consideration as perceived stress scores indicated infrequent or everyday meditation may be a stressor” (p. 16) – please elaborate on this. Has this been seen in previous research? Further discussion on these findings is warranted.

Response: this sentence was removed.

g. In the Conclusion section (p. 16), please provide additional context for these results (e.g., “This research demonstrates that many types of meditation yield varied results among adolescents in Thailand”).

Response: more context was added about these results, “This research demonstrates that meditation type yields varied mental health results among Thai boarding school adolescents.”

3. The article is presented in an intelligible fashion and is written in standard English.

a. In general, the writing can be difficult to follow and switches between past and present tense.

4. The research meets all applicable standards for the ethics of experimentation and research integrity.

a. Yes, the authors indicate the study received approval from an ethics committee and informed consent was obtained from participants.

5. The article adheres to appropriate reporting guidelines and community standards for data availability.

a. Under Data Availability, the authors state that some restrictions will apply but then below state that the data underlying the findings are fully available without restriction. Please clarify.

Response: the minimal data set has been made available.

Other notes:

6. Introduction

a. I think it would be beneficial to elaborate on mental health issues among adolescents in Asia during/after COVID-19 lockdowns, as well as any past research on meditation-based interventions in schools in Thailand.

Response: Information about adolescents in Asia has been clarified in the introduction, “In addition to increasing self-esteem, meditation has also been linked with increased resilience [8, 9], an important quality for coping with stress, anxiety, and depression, which are commonly reported mental health issues in adolescents during secondary education [10-12] and later adolescence [13]. The prevalence of these mental problems was exceptionally high amongst children and adolescents in Asia following the COVID-19 lockdown experiences [14, 15].”

b. Has past research examined the mental health benefits of Kasina, Buddha image visualization, and Manomayiddhi? If so, please include that in the introduction in t.

Response: Buddha Image and Manomayiddhi have not been previously researched, and mention of this has now been included in the Meditation Types section. Kasina has had previous mention. Please refer to the meditation style section for this information.

7. Overall

a. There seems to be a large overlap in the content of these different meditation styles, e.g. Buddha image is part of Manomayiddhi, so these results do not help us understand what components of Manomayiddhi may be helpful.

Response: This is an important point. More clarity was brought to how these styles are integrated in practice. “In the case of manomayiddhi, meditators will generally use Buddha image visualization, however; not all Buddha image meditators practice Manomayiddhi (Table 1 for practitioner totals). Buddha image visualization can be a stand-alone technique, but it is often practiced for focus prior to manomayiddhi and can be integrated within. This is similar to breathing meditation followed by vipassana and occasionally switching back to breathing meditation as necessary.”

b. There is no discussion of why some styles might not be associated with better mental health.

Response: consideration of why some styles might not be associated with better mental health has been included. “However, the integration of the aforementioned types of meditation into manomayiddhi raises questions about why recollections, vipassana, mindfulness, and kasina were not significant predictors in the regression models. Some explanation for these styles’ non-significant results may be explained by consideration of the quality of instruction. In this sample, correlation results highlight that Buddha image and manomayiddhi are practiced more frequently at Buddhist boarding schools and not at secular schools, highlighting how school types differ in relation to meditation practice. School type was also found to be a significant influence in the resilience regression model. Buddhist boarding schools are often connected to a Buddhist temple…”

Reviewer #2: Thank you for giving me the opportunity to read and comment on this paper. I find the topic very interesting. I thank the authors for their efforts at these times. Having said that, I suggested some comments which should be addressed.

1. The abstract section needs more changes, as the main results did not well articulate. Pay attention to reporting the main results of the study, as the results in the abstract do not convey the main findings of your study. The background paragraph in "Abstract" must be deleted and changed. The background section should be the shortest part of the abstract and should very briefly outline: What is already known about the subject, is related to the paper in question. What is not known about the subject and hence what the study intended to examine??

Response: An abstract which better reflects the research has been included,

“Purpose: Meditation has been demonstrated to benefit adolescent mental health. However, less-known meditation styles have received sparse research attention. This research examined various meditation styles in northern Thailand to determine which were associated with positive and negative mental health outcomes in adolescents.

Population and Methods: Boarding school students, 15-18 years, from secular and Buddhist schools in Northern Thailand provided information about meditation styles and frequency during the last month (i.e., breathing, kasina (color), Buddha image visualization, manomayiddhi, mindfulness, recollections, and vipassana). The Rosenberg Self-Esteem Scale (RSES), Resilience Inventory (RI-9), Outcome Inventory-21 (OI-21), and Perceived Stress Scale (PSS-10) were completed. Stepwise regression was used to identify the effects of meditation styles on mental health outcomes.

Results: Among 443 participants, 390 were females (87.9%). The mean age was 16.35 (0.96) years. The three most common meditation styles practiced were breathing, Buddha image visualization, and mindfulness (46.5%, 26.2%, and 22.8%, respectively). Buddha image visualization was a significant (p < .001) predictor of RSES (β =.18), RI-9 (β =.22), OI-Anxiety (β = -.23), OI-Depression (β = -.26), and PSS-10 (β = -.20), whereas manomayiddhi was a predictor of RI-9 (β= .165, p =.001), OI-Anxiety (β = -.20, p < .001), OI-Depression (β = -.18, p < .001), and PSS-10 (β= -.154, p = .002). Breathing meditation predicted OI-Depression (β= -.129, p =.004). Daily meditation frequency was associated with the best mental health scores (p <.001).

Conclusion: Buddha image visualization, manomayiddhi, and breathing meditation were predictive of adolescents' mental health. A higher practice frequency is associated with positive mental health outcomes.

2. 3. The introduction (background) here is not adequate. The background of your study did not provide context to the information discussed throughout the research paper. The background should include both important and relevant studies on meditation and it's associated with general health and mental health outcomes. This is particularly important to support your paper.

Response: This was addressed with the addition in the second paragr

---

## [Decision Letter · Decision Letter 1]

PONE-D-23-03807R1Buddha image meditation is a potent predictor for mental health outcomes: a cross-sectional study among high-school students.PLOS ONE

Dear Dr. Tinakon Wongpakaran,

Thank you for submitting your manuscript to PLOS ONE. After careful consideration, we feel that it has merit but does not fully meet PLOS ONE’s publication criteria as it currently stands. Therefore, we invite you to submit a revised version of the manuscript that addresses the points raised during the review process.

We look forward to receiving your revised manuscript.

Kind regards,

Sally Mohammed Farghaly

Academic Editor

PLOS ONE

Journal Requirements:

Reviewers' comments:

Reviewer's Responses to Questions

**Comments to the Author**

1. If the authors have adequately addressed your comments raised in a previous round of review and you feel that this manuscript is now acceptable for publication, you may indicate that here to bypass the “Comments to the Author” section, enter your conflict of interest statement in the “Confidential to Editor” section, and submit your "Accept" recommendation.

Reviewer #1: All comments have been addressed

Reviewer #3: (No Response)

2. Is the manuscript technically sound, and do the data support the conclusions?

Reviewer #1: Partly

Reviewer #3: Partly

3. Has the statistical analysis been performed appropriately and rigorously? 

Reviewer #1: Yes

Reviewer #3: Yes

4. Have the authors made all data underlying the findings in their manuscript fully available?

Reviewer #1: Yes

Reviewer #3: Yes

5. Is the manuscript presented in an intelligible fashion and written in standard English?

Reviewer #1: Yes

Reviewer #3: Yes

6. Review Comments to the Author

Reviewer #1: (No Response)

Reviewer #3: The sample size calculation seems to have deviation with the recruited subjects into the study. Please clarify.

7. PLOS authors have the option to publish the peer review history of their article (what does this mean? ). If published, this will include your full peer review and any attached files.

**Do you want your identity to be public for this peer review?** For information about this choice, including consent withdrawal, please see our Privacy Policy .

Reviewer #1: No

Reviewer #3: No

---

## [Author Response · Author response to Decision Letter 2]

19 Jul 2023

Dear Editor,

Thank you for providing us with an opportunity to improve our manuscript. Please see below our response to the reviewer

Review Comments to the Author

Reviewer #3 comment: The sample size calculation seems to have deviation with the recruited subjects into the study. Please clarify.

Author’s response: In fact, no deviation occurred. However, the author’s previous response about sample information did omit important information relevant to your follow-up.

In the author’s previous response, we wrote, “The total sample was originally calculated to accommodate SEM with a final sample of 453 participants. For this secondary data analysis, sample size calculation for linear regression was determined by the medium effect size, significance level (alpha) of 0.05, power (beta) of 0.8. This yielded a number of at least 114 required for analysis, with a final sample of 443 participants. The total sample was included for analysis.”

The total sample of 453 participants had ten outliers removed for analysis with a final total of 443 participants. This information is found in the Results section and reads, “In total, 443 responses were included in the analysis out of 453. Ten outliers were identified in the regression analysis and excluded from the final data.”

We hope our clarification would satisfy the reviewer’s question. Should you have any further questions, please let us know. Thank you so much.

Best regards,

TW, JD, and all

---

## [Decision Letter · Decision Letter 2]

PONE-D-23-03807R2Buddha image meditation is a potent predictor for mental health outcomes: a cross-sectional study among high-school students.PLOS ONE

Dear Dr. Tinakon,

Thank you for submitting your manuscript to PLOS ONE. After careful consideration, we feel that it has merit but does not fully meet PLOS ONE’s publication criteria as it currently stands. Therefore, we invite you to submit a revised version of the manuscript that addresses the points raised during the review process.

We look forward to receiving your revised manuscript.

Kind regards,

Sally Mohammed Farghaly

Academic Editor

PLOS ONE

Journal Requirements:

Reviewers' comments:

Reviewer's Responses to Questions

**Comments to the Author**

1. If the authors have adequately addressed your comments raised in a previous round of review and you feel that this manuscript is now acceptable for publication, you may indicate that here to bypass the “Comments to the Author” section, enter your conflict of interest statement in the “Confidential to Editor” section, and submit your "Accept" recommendation.

Reviewer #3: All comments have been addressed

Reviewer #4: (No Response)

2. Is the manuscript technically sound, and do the data support the conclusions?

Reviewer #3: Yes

Reviewer #4: Yes

3. Has the statistical analysis been performed appropriately and rigorously? 

Reviewer #3: Yes

Reviewer #4: Yes

4. Have the authors made all data underlying the findings in their manuscript fully available?

Reviewer #3: Yes

Reviewer #4: No

5. Is the manuscript presented in an intelligible fashion and written in standard English?

Reviewer #3: Yes

Reviewer #4: Yes

6. Review Comments to the Author

Reviewer #3: (No Response)

Reviewer #4: Reviewer’s comments

Study Title “Buddha image meditation is a potent predictor for mental health outcomes: a cross-sectional study among high-school students.”

By DeMaranville J et al.

The study reported effects of different meditation style on mental health of high school students in Thailand. After reviewing, I have some comments as follows.

Abstract

Please add the inclusion criteria for enrolling students into the study.

Introduction

-Could the authors provide more background information about the study setting. How many high school students (or general population) in Thailand practice meditation in daily life? What are the prevalence of mental health among students in boarding schools? Are they different from students who reside with biological family? Did the prevalence of mental health problems in students change during Covid-19 period?

Methods

-Could the authors describe more about inclusion criteria and recruitment process? How did those students come to join the study, if the authors did not include everyone in the school? The number of students varied from one school to another, how they were selected?

-There were six students from a secular school, would they be a good representative of students from their school?

-Did the authors explain about different types of meditation to potential study participants? Did they have a chance to learn how each type was? Were there any interventions offered before, during, or after the study period?

-How was the data collected? (by paper-based questionnaires, online, face-to-face interview, or other methods)

Results

-There were 10.6% of students who were not Buddhist, what were their religions and why they were included? If they were Christian or Muslim, they might not be comfortable looking at the Buddha mage or understand those meditation style names. How were the questions asked?

-Did the authors collect data on their lifestyle, social behaviors, or baseline mental health before meditation practice (by self-report, teacher report, or school record etc.)

7. PLOS authors have the option to publish the peer review history of their article (what does this mean? ). If published, this will include your full peer review and any attached files.

**Do you want your identity to be public for this peer review?** For information about this choice, including consent withdrawal, please see our Privacy Policy .

Reviewer #3: No

Reviewer #4: No

---

## [Author Response · Author response to Decision Letter 3]

7 Nov 2023

Dear Editor

We greatly appreciate the reviewer's keen and insightful comments, which provide us with an opportunity to refine our manuscript. Below, you will find our detailed responses to each comment.

The study reported effects of different meditation style on mental health of high school students in Thailand. After reviewing, I have some comments as follows.

Abstract

REVIEWER: Please add the inclusion criteria for enrolling students into the study.

AUTHOR: Additional information about purposive selection was added to the abstract as follows.

Participants in this study encompassed high school students who were 15-18 years old, irrespective of their gender and religious background, and were enrolled in grades 10-12 in either secular or Buddhist Thai boarding schools.

Introduction

REVIEWER:-Could the authors provide more background information about the study setting. How many high school students (or general population) in Thailand practice meditation in daily life?

AUTHOR: Yes, that information will help with contextualization. Thank you. This sentence was added on page three in the first line of the Introduction/Meditation and Mental Health section. We have added the information as follows.

Meditation is a widespread practice in Thailand that spans across all age groups. However, there is a need for more comprehensive studies to accurately gauge the prevalence of meditation among high school students and the general population in the country. Nonetheless, it is a common ritual in schools to engage in meditation and chanting before class, and it's reasonable to expect that meditation practices are more prevalent and more structured in Buddhist schools compared to secular institutions. Additional studies have indicated that approximately 17% of the general Thai population, and as much as 58% of older residents, actively participate in meditation, highlighting a significant number of practitioners with a variety of meditation styles (1, 2).

REVIEWER: What are the prevalence of mental health among students in boarding schools?

Are they different from students who reside with biological family? Did the prevalence of mental health problems in students change during Covid-19 period?

AUTHOR: Answering the above three questions has helped to contextualize the research more. Thank you. This text was added to page 3 in the middle of the “Meditation and mental health section,”

“Previous pre-pandemic research revealed parity in the mental health of students from various schools; however, loneliness issues have been reported among boarders (15, 22). Studies conducted during the Covid-19 pandemic in Southeast Asian countries, including Thailand, have indicated that Thai adolescents reported significantly higher levels of stress, anxiety, and depression symptoms compared to their counterparts in other countries (17-20). Global estimates suggest that the prevalence of depression and anxiety doubled in response to the pandemic (21).

Methods

REVIEWER: -Could the authors describe more about inclusion criteria and recruitment process? How did those students come to join the study, if the authors did not include everyone in the school?

AUTHOR: Students were invited to join via an information session. The participating schools announced the research opportunity and interested students were invited to the information session. During this session the informed consent was read, and any questions were answered. Interested students were given approximately one week to speak with their families to ask for permission to participate. Information from this explanation was added to the “Materials and Methods” section on page 6 and reads,

“This research was an observation study. The study recruited 453 participants. The Inclusion criteria for the study were:

• Students who were 15 years old and older.

• Studying in grades 10-12.

• Studying in a Thai boarding school in Northern Thailand.

• Having access to a computer with internet.

• All religious groups.

The Exclusion criteria were students with special needs and students who were blind or deaf.

In the recruitment process, students were invited to join via an information session. The participating schools announced the research opportunity and interested students were invited to the information session. During this session the informed consent was read, and any questions were answered. Interested students were given approximately one week to speak with their families to ask for permission to participate. Since participation in this study was voluntary, not every student showed interest and chose to take part. The data collection period was between July and August 2021. The schools were purposively selected to reflect similar socioeconomic status, the number of students, and the female-to-male ratio. A research assistant contacted the schools by phone and email with information about the study. Two Buddhist schools in urban areas and three secular schools in urban and suburban areas from three provinces participated. An information session was provided to interested students to learn about the research, with data collection occurring the following week for those students willing to participate. From the two Buddhist schools, 179 and 57 students were recruited, and from the three secular schools, 145, 6, and 56 students were recruited. The variation in students willing to participate was likely due to data collection occurring amidst a lockdown announcement, with many students opting to return home opposed to sheltering at their boarding school. Students with special needs and students who were blind or deaf were excluded. Written informed consent was obtained from participants and most of the participants’ guardians. This study was approved by the Research Ethics Committee, Faculty of Medicine, Chiang Mai University, Thailand. Study (code 236/2021) and approved on 10 June 2021. A related study was published elsewhere (43).

REVIEWER: The number of students varied from one school to another, how they were selected?

AUTHOR: Yes, the number of students willing and able to participate varied due to a covid-lockdown announcement that coincided with the data collection. Many students were returning home, though some remained in the schools. The selection process for students was voluntary. Additional information about this was added to the “Materials and Methods” section on page 6 and reads, “The schools were purposively selected to reflect similar socioeconomic status, the number of students, and the female-to-male ratio. A research assistant contacted the schools by phone and email with information about the study. Two Buddhist schools in urban areas and three secular schools in urban and suburban areas from three provinces participated. An information session was provided to interested students to learn about the research, with data collection occurring the following week for those students willing to participate. From the two Buddhist schools, 179 and 57 students were recruited, and from the three secular schools, 145, 6, and 56 students were recruited. The variation in students willing to participate was likely due to data collection occurring amidst a lockdown announcement, with many students opting to return home opposed to sheltering at their boarding school. Students with special needs and students who were blind or deaf were excluded.

REVIEWER-There were six students from a secular school, would they be a good representative of students from their school?

AUTHOR: The six students were willing to participate in those that remained at the school just prior to the Covid-19 lockdown. The six-student total is not intended to be representative of the school, and the authors could not justify excluding the participants due to the low number.

REVIEWER-Did the authors explain about different types of meditation to potential study participants?

AUTHOR: No explanation about meditation types was discussed. Participants were able to mark on the meditation questionnaire if they did not meditate. Names of meditation types known to be practiced were listed on the questionnaire to be marked if practiced. A space was provided to write in additional types if they were practiced.

REVIEWER: Did they have a chance to learn how each type was?

AUTHOR: No education about the different types was provided. Meditation is freely available to be practiced at temples all over Thailand and through many schools. The style of meditation listed is standard for general Thai people. This research sought to learn what meditation styles the students had practiced, and 88 % practiced at least one of those styles.

REVIEWER: Were there any interventions offered before, during, or after the study period?

AUTHOR: No interventions were provided. This research was an observation study and sought to understand which meditation styles were practiced amongst adolescents and how often during the last month.

REVIEWER:-How was the data collected? (by paper-based questionnaires, online, face-to-face interview, or other methods)

AUTHOR: Data was collected both online and with paper-based forms, depending on participants' preferences and the availability of time and equipment. Information about this is available in the Materials and Methods section and reads, “Data were collected using both electronic and paper surveys when the students were at their schools.”

Results

REVIEWER-There were 10.6% of students who were not Buddhist, what were their religions and why they were included?

AUTHOR-The 10.6% comprised minority religious groups and non-religious participants. These participants were categorized as non-Buddhist together. This population was included as these participants may still choose to meditate in their schools or at temples. A minority religious status or non-religion holder does not mean they are excluded from these practices in Thailand. Additionally, the participants were able to mark if they did not practice meditation.

REVIEWER: If they were Christian or Muslim, they might not be comfortable looking at the Buddha mage or understand those meditation style names. How were the questions asked?

AUTHOR: The form stated that this questionnaire was to be about meditation, and that participants can fill it out whether they have or have not meditated in the last month. If they had meditated, mark how frequently meditation occurred. Then the styles practiced in the last month were listed and the participants were instructed to mark any of them or to fill in additional types if relevant. Given that contemplating on a Buddha image is a common practice in Thailand, we have identified it as a separate variable. This allows individuals who meditate on images other than that of Buddha, such as non-Buddhists, to choose "other options" and provide additional details.

We have added this part in Meditation Evaluation Questionnaire (MEQ).

REVIEWER:-Did the authors collect data on their lifestyle, social behaviors, or baseline mental health before meditation practice (by self-report, teacher report, or school record etc.)

AUTHOR: Data was collected on mental health via self-report during that day. Additional questionnaires were provided regarding lifestyle and social behaviors, though that information was not included in this secondary data analysis.

No, we did not gather the data you mentioned "prior to" the participants' meditation sessions as this was not an intervention study. However, the questions related to lifestyle and social behaviors were tailored to the respective questionnaires. For instance, questions about meditation and precepts focused on the past month, while inquiries about other mental health issues pertained to the past two weeks. Additionally, their baseline mental health information was collected in the form of their past history concerning physical and mental health.

We hope our response will sufficiently address the reviewers’ concerns. We are looking forward to hearing from you soon.

Best regards,

JD, TW, NW and DW.

---

## [Decision Letter · Decision Letter 3]

PONE-D-23-03807R3Buddha image meditation is a potent predictor for mental health outcomes: a cross-sectional study among high-school students.PLOS ONE

Dear Dr. Wongpakaran,

Thank you for submitting your manuscript to PLOS ONE. After careful consideration, we feel that it has merit but does not fully meet PLOS ONE’s publication criteria as it currently stands. Therefore, we invite you to submit a revised version of the manuscript that addresses the points raised during the review process.

We look forward to receiving your revised manuscript.

Kind regards,

Sally Mohammed Farghaly

Academic Editor

PLOS ONE

Journal Requirements:

Reviewers' comments:

Reviewer's Responses to Questions

**Comments to the Author**

1. If the authors have adequately addressed your comments raised in a previous round of review and you feel that this manuscript is now acceptable for publication, you may indicate that here to bypass the “Comments to the Author” section, enter your conflict of interest statement in the “Confidential to Editor” section, and submit your "Accept" recommendation.

Reviewer #5: All comments have been addressed

2. Is the manuscript technically sound, and do the data support the conclusions?

Reviewer #5: Yes

3. Has the statistical analysis been performed appropriately and rigorously? 

Reviewer #5: Yes

4. Have the authors made all data underlying the findings in their manuscript fully available?

Reviewer #5: Yes

5. Is the manuscript presented in an intelligible fashion and written in standard English?

Reviewer #5: Yes

6. Review Comments to the Author

Reviewer #5: The authors of the paper, "Buddha image meditation is a potent predictor for mental health outcomes: a cross-sectional study among high-school students“, structured the paper nicely covering both statistical theories on a multiple linear regression model analysis and its application to examine meditation styles and mental health

outcomes. I found the paper contributes little to either statistical modeling or meditation styles and mental health outcome literature. I found some issues discussed below:

1. In the methods:

- You simply stated as you have used stepwise multivariable linear regression analysis models. You have not written the model (equation) in the method part even if the model name you wrote is not correct. How many models did you use for your data analysis? In Statistics, there is no stepwise multivariable linear regression analysis model. Please make correction on the model name. Say Multiple linear regression model analysis. I understand that for the selection of important variables/factors, you used stepwise variable selection method. Please refer materials on the model you used to assure whether you are right or not.

- How did you select two schools deliberately? No other schools? Non-probability sampling techniques such as purposive sampling, quota sampling, ...) are bias based sampling methods. It's not recommended to use such sampling methods unless a compelling situation arises.

2. In the results:

- Incorporate statistical findings, such as charts and graphs, into your content.

7. PLOS authors have the option to publish the peer review history of their article (what does this mean? ). If published, this will include your full peer review and any attached files.

**Do you want your identity to be public for this peer review?** For information about this choice, including consent withdrawal, please see our Privacy Policy .

Reviewer #5: No

---

## [Author Response · Author response to Decision Letter 4]

3 May 2024

Dear Editor, Sally Mohammed Farghaly.

We appreciate the reviewers and the editor for providing us with an opportunity to improve our manuscript. Please see below our point-by-point response to the comments.

Reviewer #5: The authors of the paper, "Buddha image meditation is a potent predictor for mental health outcomes: a cross-sectional study among high-school students”, structured the paper nicely covering both statistical theories on a multiple linear regression model analysis and its application to examine meditation styles and mental health outcomes. I found the paper contributes little to either statistical modeling or meditation styles and mental health outcome literature. I found some issues discussed below:

Response. Thank you very much for the feedback. We hope to have addressed all of the reviewer's points. For the reviewer’s information, the authors have revised one remark and included additional mention about this paper’s purpose at the end of the intro and discussion to better highlight how this paper contributes to research on meditation styles. The sentence in the intro was edited and now reads, “This study seeks to determine the relationship between different meditation styles and mental health outcomes of adolescents in northern Thailand.” Additionally, a sentence was added to the end of the discussion that reads, “Moreover, this survey of different meditation styles practiced across northern Thailand indicates some meditation types are more impactful on mental health than others. Further study on less researched meditation types, particularly death recollections, Buddha image, and manomayiddhi, are warranted.”

1. In the methods:

- You simply stated as you have used stepwise multivariable linear regression analysis models. You have not written the model (equation) in the method part even if the model name you wrote is not correct.

Response: We have added the model equation in the method part, and the incorrect titling of the method of analysis has been corrected throughout the paper (ie. abstract, methods, results).

How many models did you use for your data analysis?

Response. The number of models used was added to each of the results sections. For positive variables, “Both the self-esteem and resilience models used four iterations to arrive at the final model.” For the negative variables, “Regarding the stepwise method used, four iterations were used in the anxiety model, with three iterations used in the depression and perceived stress models.”

In Statistics, there is no stepwise multivariable linear regression analysis model. Please make correction on the model name. Say Multiple linear regression model analysis. I understand that for the selection of important variables/factors, you used stepwise variable selection method. Please refer materials on the model you used to assure whether you are right or not. Response. Thank you, this has been corrected.

- How did you select two schools deliberately? No other schools?. (non probability convenient sampling because of covid for secular schools) Non-probability sampling techniques such as purposive sampling, quota sampling, ...) are bias based sampling methods. It's not recommended to use such sampling methods unless a compelling situation arises

Response. The two Buddhist boarding schools were the only two Buddhist boarding schools in northern Thailand and both accepted. Three other secular schools also accepted. This purposive sampling method was employed in order to ensure that this study had sufficient meditators. Additionally, this study was conducted just as the Covid-19 pandemic lockdowns were going into effect. Many schools did not accept the invitation due to the pandemic, and this study accepted interested schools on a first-come first-serve basis.

2. In the results:

- Incorporate statistical findings, such as charts and graphs, into your content.

Response. Two images were included (Figure 1 and Figure 2) that depict meditation styles and meditation practice frequencies of this population.

We hope that our responses address the reviewers’ concerns. We are looking forward to receiving your feedback.

Best regards,

TW

---

## [Decision Letter · Decision Letter 4]

PONE-D-23-03807R4Buddha image meditation is a potent predictor for mental health outcomes: a cross-sectional study among high-school students.PLOS ONE

Dear Dr. Wongpakaran,

Thank you for submitting your manuscript to PLOS ONE. After careful consideration, we feel that it has merit but does not fully meet PLOS ONE’s publication criteria as it currently stands. Therefore, we invite you to submit a revised version of the manuscript that addresses the points raised during the review process.

We look forward to receiving your revised manuscript.

Kind regards,

Sally Mohammed Farghaly

Academic Editor

PLOS ONE

Journal Requirements:

Reviewers' comments:

Reviewer's Responses to Questions

**Comments to the Author**

1. If the authors have adequately addressed your comments raised in a previous round of review and you feel that this manuscript is now acceptable for publication, you may indicate that here to bypass the “Comments to the Author” section, enter your conflict of interest statement in the “Confidential to Editor” section, and submit your "Accept" recommendation.

Reviewer #6: All comments have been addressed

Reviewer #7: (No Response)

2. Is the manuscript technically sound, and do the data support the conclusions?

Reviewer #6: Yes

Reviewer #7: Yes

3. Has the statistical analysis been performed appropriately and rigorously? 

Reviewer #6: Yes

Reviewer #7: Yes

4. Have the authors made all data underlying the findings in their manuscript fully available?

Reviewer #6: Yes

Reviewer #7: No

5. Is the manuscript presented in an intelligible fashion and written in standard English?

Reviewer #6: Yes

Reviewer #7: Yes

6. Review Comments to the Author

Reviewer #6: You addressed the reviewers’ concerns. sampling technique needs to be more detail to justify your chosen method.

Reviewer #7: Review comments

General comments

I thank the Editor for the opportunity to review this manuscript which made an interesting read. Also, the findings from the study are of relevance. In reading the manuscript I made the following observations. Responding to these comments may ensure that the quality of the manuscript is improved.

1. The selection of the school was based on purposive sampling and there was no probability sampling technique in selecting the respondents. The Buddhists constituted 89.4% of the respondents. Does it mean that 10.6% of the respondents did not apply any meditation style and if so, were they included in the analysis of the data? Moreover, the Authors indicated that some respondents selected more than one meditation style. How was this taken into account in analyzing and interpreting the findings of the study.

2. The title of the study connotes a positive alignment towards ‘Buddha image meditation.’ A more expressive title should have been; Predictors of good or positive mental health outcomes among high school students in Thailand: a cross-sectional study. In any case it will be good to find out how the Authors adopted the title as presented in the manuscript.

3. Validated tools. It will be good if the Authors provide a link to each of the validated tools used in the study. The Authors reported that the Rosenberg self-esteem scale has a total score of 40. The score of the Rosenberg self-esteem scale ranges from 0-30. Authors should cross check the tool again for details.

4. An insight into the iterations as applied in the study should be provided to guide the readers on how the final regression model was obtained. Limiting the variables in the biserial correlation table may be necessary to enhance understanding of Readers.

5. There was no indication on how the sample size for the study was obtained. Authors indicated that the schools were purposively selected to reflect similar socio-economic status, the number of students and the female to male ratio. This was different from the response provided in the answer to a comment from a Reviewer. It will be important to know whether the purposive selection achieved its purpose.

6. The linear regression result in the abstract should include 95% confidence interval and p value deleted. The conclusion in the abstract section is simply a repetition of study findings and should be reviewed.

7. The sub-titles as applied in the Introduction section should be deleted. The section should be organized into distinct paragraphs. The objectives of the study should come at the end of the Introduction section of the manuscript.

8. There was no description of the study area. There was no indication on how the questionnaire was administered to the respondents especially since the Authors indicated that there were missing variables in the limitation of the study.

9. There is no need to include p values and table numbers in the discussion section of the manuscript. The subtitle named ‘limitation and future directions’ should be deleted.

10. Authors should ensure that use of references and referencing style as used in the manuscript conform to the requirements of the Journal.

7. PLOS authors have the option to publish the peer review history of their article (what does this mean? ). If published, this will include your full peer review and any attached files.

**Do you want your identity to be public for this peer review?** For information about this choice, including consent withdrawal, please see our Privacy Policy .

Reviewer #6: No

Reviewer #7: **Yes: ** EDMUND NDUDI OSSAI

---

## [Author Response · Author response to Decision Letter 5]

4 Oct 2024

Dear Editor,

We have reviewed the feedback and addressed all the comments. Additionally, we have checked the grammar and made edits for better clarity and understanding. Please see our point-by-point response to the comments below.

1. The selection of the school was based on purposive sampling and there was no probability sampling technique in selecting the respondents. The Buddhists constituted 89.4% of the respondents. Does it mean that 10.6% of the respondents did not apply any meditation style and if so, were they included in the analysis of the data? Moreover, the Authors indicated that some respondents selected more than one meditation style. How was this taken into account in analyzing and interpreting the findings of the study.

Thank you for your questions. You are correct that by purposively selection this sample was not gathered with a probability sampling technique. Information about this was added to the Limitations section and now reads, “School participation in this study lacked randomization due to purposive selection of two Buddhist boarding schools. Participants were also not randomly selected, limiting how these results can be generalized.”

Of the respondents, 10.6% marked that they did not meditate and these participants were included in the analysis. Of this sub-population of non-meditators (50 total), 38 were Buddhist religion and 12 were not Buddhist. Within the total sample (443), there were 47 participants who were non-Buddhist. It is the case that some non-Buddhist participants did meditate and some Buddhists did not meditate. All religions were included in this study. The questionnaires ask about the participant’s meditation practice in the last month. Information about the number of meditators that practice more than one style was added to the Figure 1 information about meditative styles practiced and reads as, “It was found that 132 participants practiced more than one meditation style in the last month.”

Regarding accounting for the influence of multiple meditation styles within the analysis, all meditation styles were included in the analysis. The regression results identified the strongest associations iteratively (stepwise regression) between meditation type and mental health outcomes. What may not be accounted for in the analysis is how meditation types have influenced one another (if at all). Information about this is included in the discussion section about meditation types and reads as, “Buddha image visualization can be a stand-alone technique, but it is often practiced for focus prior to manomayiddhi. This is similar to meditators who begin with breathing meditation before they practice vipassana. This interaction of meditation types suggests one type may influence another, and this is important to account for in controlled studies. It can be the case within religious schools in Thailand that Manomayiddhi is taught with a guide for novice meditators, which can elevate the quality of the practice beyond unguided meditation. It also can mean that Buddha image meditation may be guided if used prior to guided manomayiddhi. It may be that the instruction benefits Buddha image practice, and differences in outcome might be found between guided and non-guided Buddha image and manomayiddhi meditators.” Information about the influence of meditation types on one another was also added to the limitations section and reads as, “Regarding meditation practice, the participants were able to select more than one meditation practice without indicating why a given meditation was preferred over others, how frequently each meditation was practiced, and if meditation types were combined. For example, breathing meditation may be practiced for five minutes before vipassana meditation begins. That information, along with other qualitative information, would help in evaluating the accessibility, difficulty, and benefits of these meditation types.” In addition, information was added to the conclusion about mixing of meditation types in order to encourage detailed accounts of the mixing of types and influences they may have on one another. This addition reads as, “This research also highlights the complexity of observational studies about meditation particular to the use of mixing meditation techniques. Future research on meditation types should include practice frequency, detailed documentation of meditator experiences, as well as the factors associated with meditation’s effects on mental health issues.”

2. The title of the study connotes a positive alignment towards ‘Buddha image meditation.’ A more expressive title should have been; Predictors of good or positive mental health outcomes among high school students in Thailand: a cross-sectional study. In any case it will be good to find out how the Authors adopted the title as presented in the manuscript.

Thank you for questioning the use of this particular title. The title was formed in response to the study's results, which indicated that Buddha Image meditation was found to be the most influential of the meditation styles included in the model for this population. The use of ‘potent’ might be perceived as a positive alignment. However, it is more specifically based upon the observation that the Buddha Image is the most efficacious of the meditation types analyzed on mental health.

3. Validated tools. It will be good if the Authors provide a link to each of the validated tools used in the study. The Authors reported that the Rosenberg self-esteem scale has a total score of 40. The score of the Rosenberg self-esteem scale ranges from 0-30. Authors should cross check the tool again for details.

Thank you. For the RSES scoring, the original RSES did use score ranges from 0-30. Other scoring options can be used, such as 10-40. The authors have revised the text in the RSES instrument section to clarify the point that the authors of the Thai version used the 10-40 scoring option. It now reads, “The Rosenberg Self-Esteem Scale (39) is used internationally to measure self-esteem. The reliability and validity of the tool has had extensive testing in various languages and was found to be effective. A revised version of the tool was tested with Thai adolescents and found to be equally reliable but with better construct validity (40). The RSES is brief and easily administered, utilizing a Likert-scale with ten items answered on a four-point scale that range from strongly disagree (1) to strongly agree (4), with half of the items having positive wording and half having negative wording. The total scores for the Thai version range from 10 to 40 with higher scores indicating higher self-esteem. The scale was tested with this study’s participants and resulted in a Cronbach’s α of 0.80.”

4. An insight into the iterations as applied in the study should be provided to guide the readers on how the final regression model was obtained. Limiting the variables in the biserial correlation table may be necessary to enhance understanding of Readers.

Thankyou, more information was added explaining the iterative process and how many iterations the different models used. It reads as, “. Multiple linear regression model analysis using the stepwise method for selective variables was used to determine the association along with other covariates including age, sex, and school type, as they were potential confounders. Stepwise regression is an iterative process of adding and removing variables based on their statistical significance and contribution to the model.” Within the results sections, more information about how many iterations occurred was included and reads as, “Table 4 is a multiple linear regression analysis of positive mental health outcomes, having used the stepwise method for selecting variables. Both the self-esteem and resilience models used four iterations to arrive at the final model.” And “Table 5 accounts for the influence of meditative styles on negative mental health outcomes. Regarding the stepwise method used, four iterations were used in the anxiety model, with three iterations used in the depression and perceived stress models.”

Regarding the bivariate correlation chart, the authors see the wisdom of reducing the number of variables in order to help the readers understand the most relevant variables more. Within the bivariate chart, Vipassana meditation and Mindfulness meditation were removed due to limited significant associations. The Table 2 text now includes the significant associations those two meditation styles did produce and reads as, “Vipassana meditation was significantly associated with kasina meditation .199, p<.01. Mindfulness meditation was significantly associated with kasina meditation .204, p<.01 and recollection of death meditation .192, p<.01. Both Vipassana and Mindfulness meditation were not included in the correlation chart (Table 2) due to the limited number of significant results.”

5. There was no indication on how the sample size for the study was obtained. Authors indicated that the schools were purposively selected to reflect similar socio-economic status, the number of students and the female to male ratio. This was different from the response provided in the answer to a comment from a Reviewer. It will be important to know whether the purposive selection achieved its purpose.

Thank you. Information about the sample size was included and reads as, “The sample estimation was originally calculated for structural equation modeling (SEM) with a final sample of 453 participants. For this secondary data analysis, a sample size calculation for linear regression was determined by the medium effect size, significance level (alpha) of 0.05, power (beta) of 0.8. This yielded a number of at least 114 required for the analysis. The total sample of 453 was included.”

Thank you for identifying this discrepancy between responses to reviewers. The information to both reviewers was correct, but not all of it was accounted for in the manuscript. This information has now been included and reads as, “The schools were purposively selected to reflect similar socioeconomic status, the number of students, and the female-to-male ratio. Buddhist boarding schools were prioritized when contacting schools to ensure a sufficient number of meditators participated.”

6. The linear regression result in the abstract should include 95% confidence interval and p value deleted. The conclusion in the abstract section is simply a repetition of study findings and should be reviewed.

Thank you for identifying this. The authors have deleted the p-values and stated that the regression results shown were significant.

Regarding the conclusion, the authors have included more information the reader may benefit from about this study. The conclusion now reads as, “This research demonstrates that meditation type yields varied mental health results among Thai boarding school adolescents. While mindfulness, kasina, vipassanā, and recollections meditations were not significant, Buddha image visualization, manomayiddhi, and breathing meditations were found to be effective and significant predictors using both positive and negative indices of mental health. As anticipated, frequent meditation practice predicted better mental health outcomes. This research may be the first to have assessed how Buddha image and manomayiddhi influence mental health, demonstrating gaps about meditation techniques that warrant exploration in future studies. This research also highlights the complexity of observational studies about meditation particular to the use of mixing meditation techniques. Future research on meditation types should include practice frequency, more detailed documentation of meditator experiences, as well as the factors associated with meditation’s effects on mental health issues.”

7. The sub-titles as applied in the Introduction section should be deleted. The section should be organized into distinct paragraphs. The objectives of the study should come at the end of the Introduction section of the manuscript.

Thank you. Sub-titles were deleted and distinct paragraphs were formed.

8. There was no description of the study area. There was no indication on how the questionnaire was administered to the respondents especially since the Authors indicated that there were missing variables in the limitation of the study.

Thank you. Additional information about the study area and the method for administering the questionnaire was added to the methods section and now reads as, “Boarding schools in northern Thailand were purposively selected to reflect similar socioeconomic status, the number of students, and the female-to-male ratio. Buddhist boarding schools were prioritized when contacting schools to ensure a sufficient number of meditators participated. A research assistant contacted the schools by phone and email with information about the study. Two Buddhist schools in urban areas and three secular schools in urban and suburban areas from three provinces participated. In the recruitment process, participating schools announced an information session that interested students could attend to learn about the research, with data collection occurring the following week. During this session, the informed consent was read and any questions were answered. Interested students were given approximately one week to speak with their families to ask for permission to participate. Since participation in this study was voluntary, not every student showed interest and chose to take part. The data collection period was between July and August 2021. From the two Buddhist schools, 179 and 57 students were recruited, and from the three secular schools, 145, 6, and 56 students were recruited. The questionnaires were administered both online and with paper depending upon the needs of the participating schools. The variation in students willing to participate was likely due to data collection occurring amidst a lockdown announcement, with many students opting to return home as opposed to sheltering at their boarding school.”

9. There is no need to include p values and table numbers in the discussion section of the manuscript. The subtitle named ‘limitation and future directions’ should be deleted.

Thank you, the p-values have been removed. The subtitle was deleted.

10. Authors should ensure that use of references and referencing style as used in the manuscript conform to the requirements of the Journal.

Thank you. The referencing style corresponds with the journals preferred citation style Vancouver.

We hope these revisions meet the expectations of the editor and reviewers. We look forward to hearing from you soon.

Best regards,

TW and colleagues

---

## [Decision Letter · Decision Letter 5]

PONE-D-23-03807R5Buddha image meditation is a potent predictor for mental health outcomes: a cross-sectional study among high-school students.PLOS ONE

Dear Dr. Wongpakaran,

Thank you for submitting your manuscript to PLOS ONE. After careful consideration, we feel that it has merit but does not fully meet PLOS ONE’s publication criteria as it currently stands. Therefore, we invite you to submit a revised version of the manuscript that addresses the points raised during the review process.

We look forward to receiving your revised manuscript.

Kind regards,

Daniel Ahorsu, PhD

Academic Editor

PLOS ONE

Journal Requirements:

Reviewers' comments:

Reviewer's Responses to Questions

**Comments to the Author**

1. If the authors have adequately addressed your comments raised in a previous round of review and you feel that this manuscript is now acceptable for publication, you may indicate that here to bypass the “Comments to the Author” section, enter your conflict of interest statement in the “Confidential to Editor” section, and submit your "Accept" recommendation.

Reviewer #7: (No Response)

Reviewer #8: All comments have been addressed

Reviewer #9: All comments have been addressed

2. Is the manuscript technically sound, and do the data support the conclusions?

Reviewer #7: Yes

Reviewer #8: Yes

Reviewer #9: Yes

3. Has the statistical analysis been performed appropriately and rigorously? 

Reviewer #7: Yes

Reviewer #8: Yes

Reviewer #9: Yes

4. Have the authors made all data underlying the findings in their manuscript fully available?

Reviewer #7: Yes

Reviewer #8: Yes

Reviewer #9: Yes

5. Is the manuscript presented in an intelligible fashion and written in standard English?

Reviewer #7: Yes

Reviewer #8: Yes

Reviewer #9: Yes

6. Review Comments to the Author

Reviewer #7: Review comments

General comments

I commend the Authors for the responses provided to the review comments as submitted during the review of the manuscript. I am of the opinion that the manuscript has some merits. I will endeavor to point out some comments that were not taken care of from the previous review comments.

Authors should take note of the following observations;

1. The title of the study connotes a positive alignment towards ‘Buddha image meditation.’ My observation is that the findings related to Buddha image meditation remarkably influenced the writing style of the Authors and re-phrasing the title may entail almost a re-writing of the manuscript. Based on this comment, I will allow the title to be except if the Editor thinks otherwise.

2. There was no description of the study area. There was no indication of how the students were recruited into the study. (This should also be indicated in the abstract). There was no indication on how the questionnaire was administered to the respondents especially since the Authors indicated that there were missing variables in the limitation of the study.

3. There is no need to include p values and table numbers in the discussion section of the manuscript.

4. Authors should ensure that use of references and referencing style as used in the manuscript conform to the requirements of the Journal. I encourage the Authors to visit the Journal website for guidance. For example, PLOS ONE makes use of block paragraphing and also prefer square brackets for reference numbers.

5. Ethical approval. The mean age of the students was recorded as 16.35±0.96 years. However, there was no indication that ‘assent’ was obtained from the respondents before including them in the study.

6. In the comments related to study limitation, ‘randomization and randomly selected’ may not be appropriate. The point to note is that probability sampling techniques were not used in the selection of the schools and in recruiting the students into the study.

7. P values should be written as p<0.001 instead of p<.001 to avoid confusion. The table on gender should be complete bearing in mind the sample size indicated at the beginning of the table.

Reviewer #8: The paper seems to be modified based the reviewers' comments.

I just have a one comments about mother's skills and their health literacy that can have a mediated role in promoting the mental health of the school child people. So I suggest to check the following paper for more conceptualization of the mentioned issue:

Enabling mothers through improving mental health literacy and parenting skills

Biosocial Health Journal. 2024;1(1): 26-32. doi: 10.34172/bshj.4

Reviewer #9: Scientific English Translation:

This study is part of a multidisciplinary research line that is undoubtedly highly relevant in the current context. The field of mental health has expanded considerably in recent decades.

The Depth and Relevance of the Research

The in-depth analysis of the study, supported by a consistent and high-quality bibliography, reflects a solid theoretical foundation. The choice to explore the interaction between spiritual, psychological, and medical practices is innovative, given the growing emphasis on holistic approaches to health promotion, and raises issues such as the influence of cultural environment and the importance of integrative therapeutic approaches.

Furthermore, the sample used in the study is representative and large, which increases the relevance of the findings. A significant sample provides greater statistical power to the research, ensuring that the observed results have a higher degree of reliability and validity. This also facilitates the potential to generalize the conclusions to broader contexts, provided the sample's characteristics are adequately described and contextualized.

The Intersection of Spirituality, Psychology, and Medicine

One of the study’s greatest merits lies in its ability to integrate different disciplines – spirituality, psychology, and medicine – into a single body of research. This represents an integrated and holistic view of health, an approach that recognizes that human well-being is not confined to a single aspect of life. This type of research is particularly relevant, as there is an increasing movement within the health field that favors the use of therapeutic methods that consider the human being in a global sense.

In this regard, the study goes beyond merely analyzing therapeutic interventions or isolated practices, and seeks a deeper understanding of the possible interactions between different aspects of the human experience. Thus, it is understood that meditation, for example, can function not only as a spiritual practice but also as a therapeutic tool that supports emotional regulation and the development of essential psychological skills, such as mindfulness, which has been widely recognized for its benefits to mental health.

Clarity, Objectivity, and Scientific Precision

However, to further strengthen the study, some improvements can be made, particularly concerning clarity, objectivity, and scientific precision. The quality of a study largely depends on the researchers’ ability to present their findings transparently and based on solid foundations, using precise scientific language. Ambiguity in methodological terms and a lack of clarity in definitions can compromise the interpretation of results and generate uncertainties about the study’s validity. Therefore, the authors should minimize any language or expressions that might approach ambiguity or subjectivity.

In this case, it is essential to ensure that the methods and results are presented with complete transparency. Explicitly detailing the data collection process, as well as adopting valid and reliable instruments to measure the variables of interest, are crucial points that should be described objectively and supported by other studies. Similarly, the choice and proper use of statistical tools for data analysis should also be described accurately to ensure that the results are solid and supported by a robust analysis.

Inclusion and Exclusion Criteria

Another point that requires further analysis is the inclusion and exclusion criteria used to define the study’s sample. These criteria should be clearly established to avoid the introduction of possible biases or distortions in the participant selection process. The presence of any form of bias in the selection criteria can compromise the external validity of the study. Transparency in defining these criteria is essential, as it ensures that the study’s results are representative of the target population and that the conclusions are valid and applicable to different settings.

Moreover, it is important that these criteria are based on solid scientific foundations and reviewed in light of the existing literature. Including cultural, sociodemographic, and psychological factors in defining inclusion and exclusion criteria, for example, may provide a richer and more contextualized understanding of the results.

Study Objectives: To Determine or To Understand?

The formulation of the research objectives is also a point that deserves attention. The expression “This study seeks to determine the relationship between different meditation styles and mental health outcomes of adolescents in northern Thailand” could be revised to more accurately reflect what the study aims to achieve. The verb "determine" suggests a more rigid and definitive approach, while a verb such as "evaluate" or "understand" might be more appropriate, as it is common in scientific studies to seek a broader, more open understanding of the phenomena under investigation, without the intention of establishing direct and immutable causality.

The choice of terms is crucial for the study to be properly interpreted by the scientific community. Using a more flexible term such as "evaluate" or "understand" allows the researchers to maintain a level of scientific caution, acknowledging the limitations of the study and avoiding premature conclusions.

Statistical Treatment

Finally, the statistical treatment conducted in the study is a positive aspect that should be valued. The use of robust statistical analyses to validate the formulated hypotheses is essential to ensure the credibility of the results. Statistics play a crucial role in data interpretation, as they provide the necessary tools to verify the significance of findings, test hypotheses, and ensure that the results are not merely due to chance. However, the choice of statistical analyses should be carefully justified and processed, taking into account the nature of the data and the study’s objectives.

Conclusion

In summary, the study presents a valuable and innovative proposal by integrating spiritual, psychological, and medical approaches. However, greater clarity, objectivity, and scientific precision in formulating the objectives, defining inclusion and exclusion criteria, and describing the methods could further strengthen the quality of the work. The use of precise terminology and the refinement of methodological aspects contribute to ensuring the robustness of the results and the credibility of the study.

7. PLOS authors have the option to publish the peer review history of their article (what does this mean? ). If published, this will include your full peer review and any attached files.

**Do you want your identity to be public for this peer review?** For information about this choice, including consent withdrawal, please see our Privacy Policy .

Reviewer #7: **Yes: ** EDMUND NDUDI OSSAI

Reviewer #8: No

Reviewer #9: No

---

## [Author Response · Author response to Decision Letter 6]

6 Mar 2025

Review comments

General comments

I commend the Authors for the responses provided to the review comments as submitted during the review of the manuscript. I am of the opinion that the manuscript has some merits. I will endeavor to point out some comments that were not taken care of from the previous review comments.

Thank you for your helpful feedback. It was warmly received and appreciated.

Authors should take note of the following observations;

1. The title of the study connotes a positive alignment towards ‘Buddha image meditation.’ My observation is that the findings related to Buddha image meditation remarkably influenced the writing style of the Authors and re-phrasing the title may entail almost a re-writing of the manuscript. Based on this comment, I will allow the title to be except if the Editor thinks otherwise.

Thank you for sharing your perception of how the title can be interpreted. The authors’ suspect it was the use of ‘potent’ in the title that may have influenced the reviewer to view the title as positively aligned. It is not the case that ‘potent’ conveys positive alignment, only that the object (Buddha image meditation) is the strongest or most effective as an influence for the outcomes. If any other meditation type had been found to be the most effective, potent could still have been selected to convey the largest influence. The manuscript contents convey both that Buddha image meditation was the most effective predictor as well as that the meditation practices are embedded with religious influence. It is the author’s opinion that the content remains scientifically sound and educational about the cultural and religious understanding embedded within these practices.

2. There was no description of the study area. There was no indication of how the students were recruited into the study. (This should also be indicated in the abstract). There was no indication on how the questionnaire was administered to the respondents especially since the Authors indicated that there were missing variables in the limitation of the study.

Thank you for drawing the author’s attention to these points. Regarding study area, ‘northern Thailand’ is highlighted as the area of the study. However, the author’s have added the names of the provinces which the participating boarding schools were located in to provide more information (Chiang Mai, Lamphun, Uthai Thani, Phitsanulok). Regarding recruitment information and administration of the questionnaires for participants in the study, please find this in the methods section that reads, “A research assistant contacted the schools by phone and email with information about the study…. In the recruitment process, participating schools announced an information session that interested students could attend to learn about the research, with data collection occurring the following week. During this session, the informed consent was read and any questions were answered. Interested students were given approximately one week to speak with their families to ask for permission to participate. Since participation in this study was voluntary, not every student showed interest and chose to take part. The data collection period was between July and August 2021….The questionnaires were administered both online and with paper depending upon the needs of the participating schools. The variation in students willing to participate was likely due to data collection occurring amidst a lockdown announcement, with many students opting to return home as opposed to sheltering at their boarding school. Students with special needs and students who were blind or deaf were excluded. Written informed consent was obtained from participants and most of the participants’ guardians. The ethics committee approved the schools to provide permission for students whose parents were unable to be reached.” During the process of collecting data, it was found that 3% of the data was missing. This can be due to reasons such as accidentally or intentionally skipping a question. Participants were not required to complete all questions to avoid coercion for items that might make them uncomfortable. Both paper and pencil as well as online forms were used.

3. There is no need to include p values and table numbers in the discussion section of the manuscript.

Thankyou, the authors agree with this and have removed those details from the discussion.

4. Authors should ensure that use of references and referencing style as used in the manuscript conform to the requirements of the Journal. I encourage the Authors to visit the Journal website for guidance. For example, PLOS ONE makes use of block paragraphing and also prefer square brackets for reference numbers.

Thank you, the authors agree and have implemented the paragraph blocking. The in-text citation brackets are proving difficult at the moment. The authors will prepare for this change.

5. Ethical approval. The mean age of the students was recorded as 16.35±0.96 years. However, there was no indication that ‘assent’ was obtained from the respondents before including them in the study.

Information on informed consent can be found in the methods section and reads as, “…During this session, the informed consent was read and any questions were answered. Interested students were given approximately one week to speak with their families to ask for permission to participate…. Written informed consent was obtained from participants and most of the participants’ guardians. The ethics committee approved the schools to provide permission for students whose parents were unable to be reached.”

6. In the comments related to study limitation, ‘randomization and randomly selected’ may not be appropriate. The point to note is that probability sampling techniques were not used in the selection of the schools and in recruiting the students into the study.

Thank you. The author’s agree that your suggested wording has merit and have adapted the limitations section to read as, “Schools and students were recruited for the study without the use of probability sampling techniques, such as with the purposive selection of two Buddhist boarding schools. Participants also joined following announcements at participating schools, rather than by randomly selecting participants from those schools.”

7. P values should be written as p<0.001 instead of p<.001 to avoid confusion. The table on gender should be complete bearing in mind the sample size indicated at the beginning of the table.

Thank you. The authors have implemented these changes.

Reviewer #9

The formulation of the research objectives is also a point that deserves attention. The expression “This study seeks to determine the relationship between different meditation styles and mental health outcomes of adolescents in northern Thailand” could be revised to more accurately reflect what the study aims to achieve. The verb "determine" suggests a more rigid and definitive approach, while a verb such as "evaluate" or "understand" might be more appropriate

Thank you for your review of the manuscript and commentary. The authors have endeavored to improve the quality of the work along the lines of the reviewers remarks. Specifically, related to the specific wording used in the ‘objectives,’ the authors have edited the statement according to the reviewer’s remarks. The new objective reads as, “This study seeks to understand the relationship between different meditation styles and mental health outcomes of adolescents in northern Thailand.”

---

## [Decision Letter · Decision Letter 6]

PONE-D-23-03807R6Buddha image meditation is a potent predictor for mental health outcomes: a cross-sectional study among high-school students.PLOS ONE

Dear Dr. Wongpakaran,

Thank you for submitting your manuscript to PLOS ONE. After careful consideration, we feel that it has merit but does not fully meet PLOS ONE’s publication criteria as it currently stands. Therefore, we invite you to submit a revised version of the manuscript that addresses the points raised during the review process.

We look forward to receiving your revised manuscript.

Kind regards,

Daniel Ahorsu, PhD

Academic Editor

PLOS ONE

Journal Requirements:

Additional Editor Comments:

Dear Authors,

Thanks for the revision. However, it was realised that you did not respond to the comment of Reviewer 8. Therefore, I will entreat you to respond to it.

I do not know whether you may still see it therefore, I am stating it here: "The paper seems to be modified based the reviewers' comments.

I just have a one comments about mother's skills and their health literacy that can have a mediated role in promoting the mental health of the school child people. So I suggest to check the following paper for more conceptualization of the mentioned issue:

Enabling mothers through improving mental health literacy and parenting skills

Biosocial Health Journal. 2024;1(1): 26-32. doi: 10.34172/bshj.4"

Please ignore the comment of Reviewer 9 this time or any other comments from a reviewer. Only address the one above (repost from Reviewer 8).

Reviewers' comments:

Reviewer's Responses to Questions

**Comments to the Author**

1. If the authors have adequately addressed your comments raised in a previous round of review and you feel that this manuscript is now acceptable for publication, you may indicate that here to bypass the “Comments to the Author” section, enter your conflict of interest statement in the “Confidential to Editor” section, and submit your "Accept" recommendation.

Reviewer #7: (No Response)

Reviewer #9: All comments have been addressed

2. Is the manuscript technically sound, and do the data support the conclusions?

Reviewer #7: Yes

Reviewer #9: Yes

3. Has the statistical analysis been performed appropriately and rigorously? 

Reviewer #7: Yes

Reviewer #9: Yes

4. Have the authors made all data underlying the findings in their manuscript fully available?

Reviewer #7: No

Reviewer #9: Yes

5. Is the manuscript presented in an intelligible fashion and written in standard English?

Reviewer #7: Yes

Reviewer #9: Yes

6. Review Comments to the Author

Reviewer #7: Review comments

General comments

I commend the Authors for the good review of the manuscript. I am of the opinion that the manuscript should be accepted for publication.

Authors should effect these changes as presented below

1. The Title should include the country where the study was conducted

2. The mean age of the respondents should be better presented as 15.35±0.96 years

3. The Journal, PLOS ONE uses square brackets for reference numbers instead of circular brackets

Reviewer #9: This is a highly relevant and valuable study that has undergone multiple revisions, benefiting from these refinements to improve its clarity and scientific rigor, particularly concerning methodological aspects. Naturally, as with any research, there is always room for further enhancement. One potential avenue for improvement could involve a deeper exploration of the psychological mechanisms underlying meditation, particularly from a neuroscientific perspective. Understanding the neural processes at play during meditation, as well as identifying the key variables that influence its effectiveness, could provide valuable insights into how meditation can be optimized for different individuals and contexts.

Moreover, investigating the factors that contribute to a more or less beneficial meditation experience—both from the perspective of the practitioner and the technique itself—could offer a more comprehensive understanding of meditation's impact on cognitive and emotional well-being. Variables such as individual differences, meditation experience, environmental factors, and specific techniques employed could be examined in greater detail to determine their role in enhancing or diminishing the benefits of meditation. Additionally, exploring the interplay between meditation and neuroplasticity, stress regulation, and emotional resilience could further illuminate the long-term effects of meditation practices.

In this regard, the study could be further strengthened by making the introduction more explicit in its presentation of established scientific findings, rather than relying primarily on descriptive elements. A more thorough review of existing literature, incorporating recent advancements in neuroscience and psychology, could provide a stronger foundation for the study's hypotheses and methodologies. Moreover, an interdisciplinary approach, integrating insights from cognitive science, clinical psychology, and contemplative studies, could enrich the discussion and broaden the study’s implications. Additionally, a clearer articulation of how this study contributes to the broader scientific discourse on meditation would further enhance its significance and impact.

By integrating these refinements, the study would not only become more rigorous but also more insightful, paving the way for future research on the cognitive and neural mechanisms of meditation and its practical applications in promoting mental well-being. Expanding the scope to consider cultural and contextual differences in meditation practices could also provide valuable perspectives on its effectiveness across diverse populations.

7. PLOS authors have the option to publish the peer review history of their article (what does this mean? ). If published, this will include your full peer review and any attached files.

**Do you want your identity to be public for this peer review?** For information about this choice, including consent withdrawal, please see our Privacy Policy .

Reviewer #7: **Yes: ** EDMUND NDUDI OSSAI

Reviewer #9: No

---

## [Author Response · Author response to Decision Letter 7]

6 May 2025

Additional Editor Comments:

Dear Authors,

Thanks for the revision. However, it was realised that you did not respond to the comment of Reviewer 8. Therefore, I will entreat you to respond to it.

Thank you for bring this to our attention.

I do not know whether you may still see it therefore, I am stating it here: "The paper seems to be modified based the reviewers' comments.

I just have a one comments about mother's skills and their health literacy that can have a mediated role in promoting the mental health of the school child people. So I suggest to check the following paper for more conceptualization of the mentioned issue:

Enabling mothers through improving mental health literacy and parenting skills

Biosocial Health Journal. 2024;1(1): 26-32. doi: 10.34172/bshj.4"

Thank you for the recommendation.

Please ignore the comment of Reviewer 9 this time or any other comments from a reviewer. Only address the one above (repost from Reviewer 8).

Reviewers' comments:

Reviewer's Responses to Questions

Comments to the Author

1. If the authors have adequately addressed your comments raised in a previous round of review and you feel that this manuscript is now acceptable for publication, you may indicate that here to bypass the “Comments to the Author” section, enter your conflict of interest statement in the “Confidential to Editor” section, and submit your "Accept" recommendation.

Reviewer #7: (No Response)

Reviewer #9: All comments have been addressed

2. Is the manuscript technically sound, and do the data support the conclusions?

Reviewer #7: Yes

Reviewer #9: Yes

3. Has the statistical analysis been performed appropriately and rigorously?

Reviewer #7: Yes

Reviewer #9: Yes

4. Have the authors made all data underlying the findings in their manuscript fully available?

Reviewer #7: No

Reviewer #9: Yes

5. Is the manuscript presented in an intelligible fashion and written in standard English?

Reviewer #7: Yes

Reviewer #9: Yes

6. Review Comments to the Author

Reviewer #7: Review comments

General comments

I commend the Authors for the good review of the manuscript. I am of the opinion that the manuscript should be accepted for publication.

Authors should effect these changes as presented below

1. The Title should include the country where the study was conducted

Thank you. The authors have made the change.

2. The mean age of the respondents should be better presented as 16.35±0.96 years

Indeed, thank you. Change made.

3. The Journal, PLOS ONE uses square brackets for reference numbers instead of circular brackets

Acknowledged. Can this setting be changed by the copyeditor? The author is unable to make this change within Endnote despite repeated attempts.

Reviewer #9: This is a highly relevant and valuable study that has undergone multiple revisions, benefiting from these refinements to improve its clarity and scientific rigor, particularly concerning methodological aspects. Naturally, as with any research, there is always room for further enhancement. One potential avenue for improvement could involve a deeper exploration of the psychological mechanisms underlying meditation, particularly from a neuroscientific perspective. Understanding the neural processes at play during meditation, as well as identifying the key variables that influence its effectiveness, could provide valuable insights into how meditation can be optimized for different individuals and contexts.

Moreover, investigating the factors that contribute to a more or less beneficial meditation experience—both from the perspective of the practitioner and the technique itself—could offer a more comprehensive understanding of meditation's impact on cognitive and emotional well-being. Variables such as individual differences, meditation experience, environmental factors, and specific techniques employed could be examined in greater detail to determine their role in enhancing or diminishing the benefits of meditation. Additionally, exploring the interplay between meditation and neuroplasticity, stress regulation, and emotional resilience could further illuminate the long-term effects of meditation practices.

In this regard, the study could be further strengthened by making the introduction more explicit in its presentation of established scientific findings, rather than relying primarily on descriptive elements. A more thorough review of existing literature, incorporating recent advancements in neuroscience and psychology, could provide a stronger foundation for the study's hypotheses and methodologies. Moreover, an interdisciplinary approach, integrating insights from cognitive science, clinical psychology, and contemplative studies, could enrich the discussion and broaden the study’s implications. Additionally, a clearer articulation of how this study contributes to the broader scientific discourse on meditation would further enhance its significance and impact.

By integrating these refinements, the study would not only become more rigorous but also more insightful, paving the way for future research on the cognitive and neural mechanisms of meditation and its practical applications in promoting mental well-being. Expanding the scope to consider cultural and contextual differences in meditation practices could also provide valuable perspectives on its effectiveness across diverse populations.

Thank you for these many thoughtful recommendations. Regarding the biological and neural correlates of meditation, information was added in the early part of the introduction that reads as follows: “Neurobiological research has found that meditation may influence mental health through changes in the brain’s structure and its functioning. Mindfulness meditation is associated with increased activation in the prefrontal cortex and decreased activity in the amygdala (16, 17). These changes may support cognitive control and emotional reactivity, which may help with emotional regulation, resilience, and positive mental health outcomes.”

The authors have more clearly articulated how this research contributes to the conversation surrounding meditation. This addition can be found in the paragraph before “Materials and Methods,” it reads, “Mindfulness meditation has received considerable research attention, with much less research investigating multiple meditation styles within the same population. Thai people practice a wide array of meditation types embedded in cultural contexts that influence their meditation practice. This study expands the conversation about meditation by accounting for mindfulness meditation as well as lesser-known styles such as Buddha Image and Manomayiddhi. Adolescent mental health in Thailand can then be assessed in consideration of the many meditation types they may practice, as well as by accounting for their frequency of practice and sociodemographic influences.”

7. PLOS authors have the option to publish the peer review history of their article (what does this mean?). If published, this will include your full peer review and any attached files.

Do you want your identity to be public for this peer review? For information about this choice, including consent withdrawal, please see our Privacy Policy.

Reviewer #7: Yes: EDMUND NDUDI OSSAI

Reviewer #9: No

---

## [Decision Letter · Decision Letter 7]

PONE-D-23-03807R7Buddha image meditation is a potent predictor for mental health outcomes: a cross-sectional study among Thai high-school studentsPLOS ONE

Dear Dr. Wongpakaran,

Thank you for submitting your manuscript to PLOS ONE. After careful consideration, we feel that it has merit but does not fully meet PLOS ONE’s publication criteria as it currently stands. Therefore, we invite you to submit a revised version of the manuscript that addresses the points raised during the review process.

We look forward to receiving your revised manuscript.

Kind regards,

Daniel Ahorsu, PhD

Academic Editor

PLOS ONE

Journal Requirements:

Additional Editor Comments:

Dear Authors,

Thanks for the revision.

This was my earlier comment to you but there has not been a revision or rebuttal to the comment (actually from reviewer 8). I realised that you did not respond to the comment of Reviewer 8. Therefore, I will entreat you to respond to it.

I do not know whether you may still see it therefore, I am stating it here: "The paper seems to be modified based the reviewers' comments.

I just have a one comments about mother's skills and their health literacy that can have a mediated role in promoting the mental health of the school child people. So I suggest to check the following paper for more conceptualization of the mentioned issue:

Enabling mothers through improving mental health literacy and parenting skills

Biosocial Health Journal. 2024;1(1): 26-32. doi: 10.34172/bshj.4"

Please ignore the comment of Reviewer 9 this time or any other comments from a reviewer. Only address the one above (repost from Reviewer 8).

Reviewers' comments:

Reviewer's Responses to Questions

**Comments to the Author**

1. If the authors have adequately addressed your comments raised in a previous round of review and you feel that this manuscript is now acceptable for publication, you may indicate that here to bypass the “Comments to the Author” section, enter your conflict of interest statement in the “Confidential to Editor” section, and submit your "Accept" recommendation.

Reviewer #7: All comments have been addressed

2. Is the manuscript technically sound, and do the data support the conclusions?

Reviewer #7: Yes

3. Has the statistical analysis been performed appropriately and rigorously? 

Reviewer #7: Yes

4. Have the authors made all data underlying the findings in their manuscript fully available?

Reviewer #7: No

5. Is the manuscript presented in an intelligible fashion and written in standard English?

Reviewer #7: Yes

6. Review Comments to the Author

Reviewer #7: Review comments

General comments: I commend the Authors for their good understanding all through the period of the review of this manuscript. I recommend that the manuscript should be accepted for publication.

It is important that the use of square brackets for reference numbers is corrected before publication.

7. PLOS authors have the option to publish the peer review history of their article (what does this mean? ). If published, this will include your full peer review and any attached files.

**Do you want your identity to be public for this peer review?** For information about this choice, including consent withdrawal, please see our Privacy Policy .

Reviewer #7: **Yes: ** EDMUND NDUDI OSSAI

---

## [Author Response · Author response to Decision Letter 8]

28 May 2025

Dear Editor,

Based on the editor’s comments, “This was my earlier comment to you but there has not been a revision or rebuttal to the comment (actually from reviewer 8). I realised that you did not respond to the comment of Reviewer 8. Therefore, I will entreat you to respond to it.

I do not know whether you may still see it therefore, I am stating it here: "The paper seems to be modified based the reviewers' comments.

I just have a one comments about mother's skills and their health literacy that can have a mediated role in promoting the mental health of the school child people. So I suggest to check the following paper for more conceptualization of the mentioned issue:

Enabling mothers through improving mental health literacy and parenting skills

Biosocial Health Journal. 2024;1(1): 26-32. doi: 10.34172/bshj.4"

Please ignore the comment of Reviewer 9 this time or any other comments from a reviewer. Only address the one above (repost from Reviewer 8).”

The editor's remarks about disregarding the reviewer’s comments and considering the use of an unrelated manuscript on mental health literacy and parenting skills were already addressed in the last round of reviews. It is the second time the editor has sent that reply. Please consider that matter sufficiently addressed.

Comments to the Author

Reviewer #7: Review comments

General comments: I commend the Authors for their good understanding all through the period of the review of this manuscript. I recommend that the manuscript should be accepted for publication.

It is important that the use of square brackets for reference numbers is corrected before publication.

Thank you for your commendation and for recommending the acceptance of the manuscript. The square brackets have been added.

We believe we have addressed all comments. If there are any additional points for consideration, please let us know."

Best regards,

TW

---

## [Editor Report · Decision Letter 8]

Buddha image meditation is a potent predictor for mental health outcomes: a cross-sectional study among Thai high-school students

PONE-D-23-03807R8

Dear Dr. Wongpakaran,

We’re pleased to inform you that your manuscript has been judged scientifically suitable for publication and will be formally accepted for publication once it meets all outstanding technical requirements.

Kind regards,

Daniel Ahorsu, PhD

Academic Editor

PLOS ONE
---

## [Editor Report · Acceptance letter]

PONE-D-23-03807R8

PLOS ONE

Dear Dr. Wongpakaran,

I'm pleased to inform you that your manuscript has been deemed suitable for publication in PLOS ONE. Congratulations! Your manuscript is now being handed over to our production team.

Kind regards,

on behalf of

Dr. Daniel Ahorsu

Academic Editor

PLOS ONE